

# Exploring the potential of forest snow modelling at the tree and snowpack layer scale

Giulia Mazzotti[1], Jari-Pekka Nousu[1,2,3], Vincent Vionnet[4], Tobias Jonas[5], Rafife Nheili[1], Matthieu Lafaysse[1]

[1]Univ. Grenoble Alpes, Université de Toulouse, Météo-France, CNRS, CNRM, Centre d'Études de la Neige, Grenoble, France
[2]Water, Energy and Environmental Engineering Research Unit, University of Oulu, Oulu, Finland
[3]Bioeconomy and Environment, Natural Resources Institute Finland, Helsinki, Finland
[4]Meteorological Research Division, Environment and Climate Change Canada, Dorval, QC, Canada
[5]WSL Institute for Snow and Avalanche Research SLF, Davos, Switzerland

*Correspondence to*: Giulia Mazzotti (giulia.mazzotti@meteo.fr)

**Abstract.** Boreal and subalpine forests host seasonal snow for multiple months per year, however snow regimes in these environments are rapidly changing due to rising temperatures and forest disturbances. Accurate prediction of forest snow dynamics, relevant for ecohydrology, biogeochemistry, cryosphere, and climate sciences, requires process-based models. While snow schemes that track the microstructure of individual snow layers have been proposed for avalanche research, tree-scale process resolving canopy representations so far only exist in a few snow-hydrological models. A framework that enables layer and microstructure resolving forest snow simulations at the meter scale is lacking to date. To fill this research gap, this study introduces the forest snow modelling framework FSMCRO, which combines two detailed, state-of-the art model components: the canopy representation from the Flexible Snow Model (FSM2), and the snowpack representation of the Crocus ensemble model system (ESCROC). We apply FSMCRO to discontinuous forests at boreal and subalpine sites to showcase how tree-scale forest snow processes affect layer-scale snowpack properties. Simulations at contrasting locations reveal marked differences in stratigraphy throughout the winter. These arise due to different prevailing processes at under-canopy versus gap locations, and due to variability in snow metamorphism dictated by a spatially variable snowpack energy balance. Ensemble simulations allow us to assess the robustness and uncertainties of simulated stratigraphy. Spatially explicit simulations unravel the dependencies of snowpack properties on canopy structure at a previously unfeasible level of detail. Our findings thus demonstrate how hyper-resolution forest snow simulations can complement observational approaches to improve our understanding of forest snow dynamics, highlighting the potential of such models as research tool in interdisciplinary studies.

## 1 Introduction

Seasonal snow takes many roles in the Earth's systems: as part of the land surface, it acts as reflective and insulating material, substantially influencing the Earth's energy budget (e.g., Thackeray and Fletcher, 2016; Colman, 2013; Sturm et



al., 1997); it constitutes an important water storage, shaping the hydrograph of snow-dominated catchments (e.g., Barnhart et al., 2016; Bales et al., 2006; Viviroli and Weingartner, 2004); it is a crucial ecosystem and habitat component, affecting animal movement, food accessibility, and soil thermodynamics and biogeochemistry (e.g., Boelman et al., 2019; Gilbert et

al., 2017; Stark et al., 2020; Zhang et al., 2018); and it can lead to natural hazard (Schweizer et al., 2003). As Northern Hemisphere seasonal snow often occurs in forested areas (Rutter et al., 2009; Kim et al., 2017), the dynamics of the forest snow cover are relevant in any of these contexts, which is why process-based (or physics-based) models applied across disciplines need to accurately capture the processes that shape forest snow cover evolution. Today, both seasonal snow regimes and forest structure are subject to rapid change (e.g., Notarnicola, 2020; Mote et al., 2018; Goeking and Tarboton,

2020; Seidl et al., 2017). Process-based models are our best available tool for predicting the evolution of forested snow-covered environments under unprecedented conditions, enabling us to assess the impacts of ongoing change.

Plenty of process-based models have evolved in the context of different research disciplines, with widely varying complexity largely determined by the intended application (e.g., Etchevers et al., 2004; Essery et al., 2009; Krinner et al., 2018). For instance, snow cover presence or absence is the key variable of interest for land surface models that primarily target accurate

simulation of land-atmosphere energy exchanges; hydrological models require snow water equivalent to be well quantified; and avalanche risk estimates rely on information on the microstructure of individual snowpack layers. Consequently, snowpack and canopy representations in hydrological, atmospheric, and land surface models, feature a broad range of model structures, where the capability to explicitly represent more processes, and in a more detailed manner, is usually linked to higher model complexity.

Snowpack properties and internal processes can exhibit strong vertical variability arising from the history of individual snow layers. Resolving the physical properties of individual layers requires complex, multi-layer snow physics models capable of prognostically tracking the evolution of snow's microstructural and thermal states, including variables such as temperature, liquid water content, density, and snow microstructure descriptors. The development of such models has traditionally been driven by avalanche research (e.g., Morin et al., 2020; Bartelt and Lehning, 2002; Vionnet et al., 2012). More recently, these

models have found further use in the remote sensing community, due to the need for a-priori knowledge on snow physical properties when interpreting electromagnetic signals (e.g., Picard et al., 2018; Kontu et al., 2017; Picard et al., 2022). In contrast, the use of detailed snowpack schemes to study the snowpack's influence on terrestrial processes is less common. Applications include permafrost (Barrere et al., 2017; Gouttevin et al., 2018) and shrub tundra thermal regimes (Domine et al., 2016), but only rarely ecological research (Saccone et al. 2013; Domine et al., 2018; Ouellet et al., 2017) and forest areas

(Rasmus et al., 2016). When applied to forests, these models are usually coupled to big-leaf canopy representations intended for coarse-resolution simulations (Gouttevin et al., 2018; Nousu et al., 2023), which hampers our understanding of the interactions between canopy and snowpack layering at small spatial scales.

Indeed, the canopy's impact on snowpack mass and energy fluxes is dictated by tree-scale processes and thus highly variable in space, creating strong horizontal snowpack heterogeneity (Safa et al., 2021; Trujillo et al., 2007; Mazzotti et al., 2019). In

recent years, approaches to explicitly resolve this variability have been brought forward by the snow-hydrological



community to meet the need for accurate snowmelt estimates from forested watersheds for downstream water provision (Bales et al., 2006), improved understanding of the role of canopy gaps in snow cover retention (Ellis et al., 2013; Broxton et al., 2020), and to inform forest management practices in support of sustainable water management (Krogh et al., 2020). Leveraging the opportunities offered by novel observational systems and increasingly detailed canopy structure datasets,

these efforts have led to hyper-resolution models that can explicitly resolve canopy structure variability at the meter-scale (Broxton et al., 2015; Mazzotti et al., 2020b,c). Current implementations and applications range from case studies at the scale of few km² (Mazzotti et al., 2023) to entire catchments spanning of km² (Lewis et al., 2023, Moeser et al., 2020) as well as operational modelling at the national scale (Mott et al., 2023). In these cases, however, the detailed canopy representations are coupled to intermediate-complexity snow schemes that represent few layers only, without

parametrization of microstructure.

A modelling system that combines both, the most complex snow and canopy representation, is lacking to date. At the same time, a few observational studies have provided evidence that tree-scale processes do impact layer-scale snowpack properties (Bouchard et al., 2022; Teich et al., 2019; Molotch et al., 2016). These observations document, for example, different microstructural properties under trees, in the unloading zone, and in canopy gaps. Increased accuracy in representing how

such small-scale canopy processes impact snowpack features would enable new forest snow model applications. For instance, it remains unexplored how snowpack heterogeneity and forest structure interact to shape ecologically relevant snow properties and resultant exchange processes between soil, snow, and atmosphere (Lembrechts et al., 2019; Bramer et al., 2018; Martz et al., 2016). Assessing the influence of forest management strategies on snowpack layering, as well as cascading impacts on ecology and biogeochemistry, would require such a complex model.

To fill this research gap, we here combine tree-scale canopy structure and layer-scale physics-based snowpack representations into one modelling framework. Our system merges capabilities of the canopy implementation of the Flexible Snow Model, FSM2, (Mazzotti et al., 2020b,c) and the Ensemble System Crocus snowpack model, ESCROC, (Lafaysse et al., 2017), creating a framework that 1) extends current applications of layer- and microstructure resolving snowpack modelling to heterogeneous forest environments, 2) enhances tree-scale process resolving forest snow simulations by the

capability to represent snow microstructure at increased vertical resolution, and 3) includes a notion of uncertainties associated with the snowpack representation in terms of an ensemble. As a proof of concept, we then apply the model to boreal and sub-alpine forests, providing first evidence on how the spatiotemporal dynamics of forest snow processes translates to snowpack properties. Specific goals of our study are thus 1) to introduce the modelling system; 2) to present a first scientific application, showcasing the model's potential as a research tool; and 3) to identify potential model

improvements and motivate possible ways forward. Ultimately, we hope to inspire and provide guidance for new research and applications using novel hyper-resolution forest snow modelling tools.

This paper is structured around our objectives as follows: in the methods section, we introduce the modeling framework and present the sites at which it was tested. In the results section, we first assess the plausibility of our simulations and then present ways to analyze the wealth of information contained in the simulations, aimed at exploring canopy structure impacts



on internal snowpack properties. Based on these results, we finally discuss the system, our findings and potential future developments and applications.

## 2 Methods

### 2.1 Modelling Framework

To enable simulations that resolve canopy-snow interactions at the scale of individual trees and internal snowpack processes
at the scale of individual physical layers, the modelling framework used in this study combines elements of two state-of the art models: Crocus and FSM2. The choice of these two models was motivated by their individual capabilities, their current widespread use in operational as well as research applications, and it leveraged the authors' involvement in their developments. The two models and their integration into a single framework are outlined in the following.

#### 2.1.1 The Crocus Snowpack Model and ESCROC ensemble system

The Crocus snowpack model (Brun et al., 1989, 1992) is a physics-based snow model of high complexity originally developed in the context of avalanche hazard forecasting. It represents snowpack layering with a Lagrangian, dynamic discretization, i.e., one or several snow layers are created upon a snow precipitation event but can be merged once they attain sufficiently similar physical properties. The model represents the major internal snowpack processes, including heat diffusion, compaction, liquid water transport, and snow metamorphism. Each layer is characterized by the state variables
depth, density, liquid water content, temperature, age, and microstructure descriptors. Snow type can be diagnosed from these snow microstructure variables. Their evolution in time depends on temperature, temperature gradients, and liquid water content (Morin et al., 2013), which evolve depending on surface energy and mass balance terms computed by the model, and parameterizations of physical properties of fresh snow.

Crocus is integrated in the ISBA land surface model within the SURFEX system (Masson et al., 2013), and therefore
coupled to a soil scheme (Decharme et al., 2011). A detailed description of the Crocus snow scheme is provided by Vionnet et al. (2012). Since then, there have been several updates and enhancements to the model (Carmagnola et al., 2014; Tuzet et al., 2017). Most important, Lafaysse et al. (2017) extended Crocus by additional parametrizations for a variety of snow processes to yield a multi-physics ensemble modeling framework (ESCROC, Ensemble System CROCus). The ensemble spread provides an estimate of model uncertainty arising from uncertainty in the process parameterizations.

Crocus has seen widespread usage and continued development, for both research purposes (Dumont et al., 2020; Di Mauro et al., 2019; Spandre et al., 2019) and operational use at MétéoFrance (Le Moigne et al., 2020; Vernay et al., 2022). In response to repeated interest from the snow modelling community to incorporate elements of Crocus in other modelling systems, e.g. CryoGrid (Zweigel et al., 2021), WRF-Hydro (Eidhammer et al., 2020), or the whole code as in MAR (Gallée et al., 2001, Navari et al., 2021), a standalone version of Crocus was recently established. It allows for Crocus to be more easily
implemented within existing land surface models, which has to date been achieved in SVS2 (Vionnet et al., 2022).





Within SURFEX, Crocus can be coupled to an explicit canopy representation for forest simulations. MEB, the corresponding scheme ('Multiple Energy Balances', Boone et al., 2017), follows a big-leaf approach, where canopy structure is characterized by specifying vegetation class, Leaf Area Index (LAI), and canopy height. Due to intricate dependencies with other components of SURFEX, including MEB was beyond the scope of the standalone Crocus version. For the same reason,

adapting MEB to represent tree-scale processes is nontrivial. So far, the use of Crocus-MEB is limited to few site-scale studies (Vincent et al., 2018; Nousu et al., 2023).

### 2.1.2 The Flexible Snow Model FSM2 and hyper-resolution canopy representation

The Flexible Snow Model, FSM2, (Mazzotti et al., 2020b,c) is an intermediate-complexity snow model evolving from the Factorial Snow Model, FSM, (Essery, 2015) and adapted for high resolution (meter-scale) simulations in forested areas.

Originally, FSM was developed to investigate the performance of snow schemes used in LSMs, and aimed to provide a platform that would allow easy integration and testing of alternative snow properties and processes parametrizations (Essery et al., 2013). The snowpack in FSM/FSM2 is thus represented with a few layers only (three by default) and not aimed at capturing physical layers. The FSM2 canopy structure is represented as one model layer, which is coupled to the snowpack (surface) via a canopy air space, and the representation of canopy-snow interactions is based on established parametrizations.

However, radiation transfer through the local three-dimensional canopy structure may be explicitly resolved by an external radiation transfer model (such as HPEval, Jonas et al., 2020) associated with FSM hyper-resolution runs.

Mazzotti et al. (2020b) presented a version specifically intended for the simulation of forest snow cover at spatial resolutions of just a few meters (FSM2.0.3). For a detailed description, we refer to Mazzotti et al. (2020b,c) and Mazzotti et al. (2023). The main difference between this hyper-resolution version and the default canopy representation is that it uses a diverse set

of process-specific canopy structure descriptors, allowing different processes to be affected by different and potentially uncorrelated local canopy features (e.g., a forest gap can at the same time experience little interception but frequent shading). Canopy descriptors computed at each modelled location capture its structural diversity with horizontal, vertical, local, and stand-scale metrics. As stated above, transmission of direct shortwave radiation through the canopy, which is dictated by the presence of canopy elements in the path of the solar beam and thus highly variable in time and space, is not parametrized.

Instead, FSM2 accepts transmissivity time series as model input, which can be obtained from any external radiative transfer model. In doing so, FSM2 maintains a simple model structure while leveraging the accuracy of radiative transfer models that resolve canopy shortwave radiation transmission explicitly.

FSM2 has been shown to replicate spatiotemporal forest snow distribution patterns well at boreal and subalpine sites (Mazzotti et al., 2023, 2020b). The model has so far been used for research purposes (Mazzotti et al., 2023) and as the

starting point for the development of intermediate-resolution modelling strategies (Mazzotti et al., 2021b) that are today implemented in the modelling framework of Switzerland's Operational Snow-Hydrological Service (Mott et al., 2023).

Because FSM2 was developed as a snow research model and not designed to be coupled with atmospheric models over all kinds of land surfaces, its code is much lighter than that of a typical LSM (such as SURFEX). Both the canopy and the soil



representations comprise only the state variables that are relevant to their interactions with the snowpack. The simplicity of
the code, however, makes installation and usage relatively straightforward. New model developers can familiarize with the
entire code rather quickly, and model enhancements are generally easier to implement than in a typical LSM. This is
certainly a reason why FSM/FSM2 has become a popular model tool in snow (hydrology) research (e.g., Magnusson et al.,
2019; Smyth et al., 2022; Alonso-González et al., 2022; Rutter et al., 2023).

### 2.1.3 FSMCRO: integration of the FSM2 canopy and ESCROC snow representations

Technically, the integration of the two models was achieved by incorporating the Crocus snowpack scheme into the FSM2
codebase as alternative snow scheme, which is in line with the purpose of the standalone Crocus version. The integration
was facilitated by the fact that both models are coded in Fortran and that driving data required by Crocus largely corresponds
to that used by FSM2 (i.e., including snow- and rainfall rates, short- and longwave radiation fluxes, air temperature and
pressure, relative humidity, and wind speed). As further advantage, this implementation preserved the simple structure of the
FSM2 code. The current implementation also enables version-tracking of both model components separately, which will
facilitate continued development in the future and avoid long-term code divergence between the different implementations of
Crocus in SURFEX, FSM2, and SVS2.

In most LSMs that represent canopy and snowpack as separate model layers, numerical coupling entails the sequential
computation of first the canopy energy balance, including the energy fluxes to the snowpack, and then heat diffusion through
the snowpack, with the previously computed energy fluxes as boundary condition (e.g., Boone et al., 2017, Lawrence et al.
2019). Snow surface temperature affects (and is potentially updated during) both steps, which can lead to numerical
inconsistencies and instabilities if solving of the two steps occurs separately. While both FSM2 and MEB-Crocus have this
structure, their approach to avoid numerical instabilities differs (see Essery (2015) for FSM, as well as Appendix I in Boone
et al., (2017) for MEB-Crocus). It must further be noted that the consideration of snow surface temperature in the coupling
between snow surface fluxes and internal snowpack energy budget (Fourteau et al., 2023) does not follow the same approach
between FSM/FSM2 (skin temperature) and Crocus (first layer temperature). Initial tests using both the skin and the first-
layer approaches revealed that direct coupling of the FSM2 canopy implementation with Crocus was prone to instabilities,
particularly in case of very thin snow surface layers.





**Figure 1: Overview of the FSMCRO modelling framework and its components, as well as of the Sodankylä (blue) and Laret (green) sites, including their location in Finland and Switzerland, the canopy height model, the location of snow depth survey plots, and site pictures. Parts of this figure are adapted from Mazzotti et al. (2021b).**

To circumvent the issue, and to enable the use of Crocus without further modification, we opted for a 'zero- layer' approach, in which FSM provide subcanopy forcings, but remains uncoupled to Crocus. Rather than using the FSM2 subroutine that solves the canopy energy balance and computes energy fluxes at the snow surface, we implemented meteorological transfer functions by which the above-canopy meteorological data is modified to represent sub-canopy meteorological states, as illustrated in the schematic in Figure 1. These sub-canopy meteorological states are then fed to Crocus, which is then applied in the same way as for an open site.

Modifications applied to the above-canopy meteorological data are based on the forest snow process implementations in FSM2, the main goal being an accurate representation of the spatio-temporal variability of sub-canopy micrometeorological conditions as dictated by canopy structure. This also involves the treatment of canopy snow interception and its subsequent depletion as modifications to the precipitation input. Although this approach sacrifices some features of an explicit and



coupled canopy, it maintains the main conceptual assets of FSM2 in terms of the inclusion of detailed, process-specific canopy structure metrics and time-varying transmissivity for direct shortwave radiation. Consequently, it accounts for the

different spatiotemporal patterns of the different meteorological variables which are at the core of the FSM2 canopy implementation. The equations used are reported in the Appendix A1. Advantages and limitations of this implementation are discussed in section 4.1.

## 2.2 Model application: sites, datasets, and simulations

For a first application of FSMCRO, we leverage datasets from the two forest sites described in Mazzotti et al. 2020b and

2021b, which include a boreal and a subalpine site and were used in the context of FSM2 development. The following sections describe the sites and the data sources and provide an overview of the simulations. Location of and canopy structure at the sites are shown in Figure 1. The use of published datasets (Mazzotti et al., 2020a, 2021a) allowed us to focus on the novelties of FSMCRO and ensured comparability with results obtained with earlier FSM2 simulations.

### 2.2.1 Sodankylä, Finland

As example of a subarctic boreal forest site, we applied the model to forest locations within the boundaries of the Finnish Meteorological Institute Arctic Research Centre at Sodankylä. located at 67°22'N, 26°38'E, and 179 m a.s.l. During a field campaign in April 2019, spatially distributed snow depth observations at ~2m spacing along forest transects were conducted on a 120 x 80 m area in a discontinuous Scots pine forest. Mazzotti et al. (2020b) used the snow depth observations to validate snow distribution patterns simulated by FSM2. Hemispherical images co-registered with snow depth observations to

were used to obtain sky-view fraction time series of canopy transmissivity for direct shortwave radiation at each surveyed location, by applying the radiative transfer model HPEval (Jonas et al., 2020). All other necessary canopy structure metrics were derived from a canopy height model at 1m resolution, which was based on terrestrial laser scanning data acquired during the same campaign. All meteorological forcings necessary to drive FSM2 were measured by an automatic weather station on site and assembled by Mazzotti et al. (2020b).

### 225 2.2.2 Laret, Switzerland

Our subalpine forest site is located at Laret, 46°50'N, 9°52'E and 1520 m a.s.l., at 4-km distance from the WSL Institute for Snow and Avalanche Research SLF in Davos. We used snow depth observations acquired during peak accumulation and throughout the snowmelt phase of water year 2019 at three 50 x 50 m plots, along a north- and a south- exposed edge and within closed canopy of a Norway spruce forest, with an equivalent setup as the Sodankylä data. Likewise, canopy structure

metrics and time series of transmissivity for direct shortwave radiation were derived from a canopy height model at 1-m resolution (based on a helicopter-borne lidar acquisition in 2010) and from co-registered hemispherical images subsequently analyzed with HPEval. Meteorological forcing data was available from an on-site automatic weather station operated by SLF, further detail is provided by Mazzotti et al. (2020b).





To complement the point locations of the manual measurements, we further consider the full extent of the 250m x 400m area
of discontinuous forest at Laret shown in Figure 1 (canopy height model), which is located within the study domain of
Mazzotti et al. (2021b). In their study, they derived canopy structure input for FSM2 at 2-m spacing to enable fully
distributed simulations at 2-m resolution. This included calculation of synthetic hemispherical images for the subsequent
creation of datasets of time-varying transmissivity for direct shortwave radiation and sky-view fraction, based on the
methodology presented by Webster et al. (2020). These datasets were leveraged in this study to achieve fully distributed
FSMCRO simulations.

### 2.2.3 Overview of FSMCRO simulations

Three sets of FSMCRO simulations were considered in this study:

1. Deterministic point simulations performed at the snow survey sites from Mazzotti et al. 2020b, designed to capture a
   broad range of forest structures, including sites in both Sodankylä and Laret. These simulations served to assess
whether snow distribution captured by FSMCRO was consistent with manual snow depth observations, and for a
   general assessment of differences in simulated snow stratigraphy at locations with varying canopy structure.
2. Spatially explicit, deterministic simulations over a 250 x 400 m area in Laret at 2 m spatial resolution, aimed at
   showcasing the fully resolved spatiotemporal variability of snowpack properties and their link to canopy structure.
3. Ensemble simulations at a subset of the snow survey sites, including i) points along a ~100 m transect across a forest
gap within the Sodankylä site, intended to assess the robustness of simulated differences between locations with
   contrasting canopy structure compared to the uncertainties of the snow model parameterizations, and ii) a point
   featuring canopy properties that represent a spatial average of the spatially explicit model domain in Laret, aimed at
   contrasting the fully resolved variability captured by the spatially explicit simulation to the ensemble spread at the
   point with average forest properties.

The deterministic simulation applies the Crocus options currently used operationally at MétéoFrance. The ensemble used
here comprises 35 members and is based on the E2 ensemble introduced in Lafaysse et al. (2017). As only difference, the
snow metamorphism options based on Flanner and Zehnder (2006), which is not yet available in the standalone version of
Crocus, were replaced by an improved metamorphism parametrization recently developed at MétéoFrance ('B21',
unpublished but used e.g. in Dick et al., 2023). The Crocus options used for these runs are listed in the Appendix A2.
All simulations were run from October 1st, 2018, to May 31st, 2019, corresponding to the period covered by the
meteorological forcing assembled by Mazzotti et al. (2020b), with a temporal resolution of one hour. The model was
initialized with homogeneous soil temperature conditions (FSM2 default: 285K) and no snow.



## 3 Results

Simulations obtained with FSMCRO cover five dimensions: 1) different snowpack properties; 2) their vertical distribution
within the snow profile; 3) their horizontal variability across modelled locations; 4) their evolution in time; and 5) their
uncertainty as captured by the spread of the ensemble. The following sections present some ideas on how this wealth of
information can be exploited, depending on what dimensions are of primary interest to the analysis. Following some
plausibility considerations, we assess the different snow states and the seasonal evolution of their vertical profiles at two
contrasting locations. We then take a closer look at how canopy structure controls the horizontal variability of snow
stratigraphy by analyzing simulations along a discontinuous forest transect and contrast this variability to associated model
uncertainties. Finally, we show how FSMCRO simulations can be applied to estimate sub-grid variability of snow surface
properties. While by no means exhaustive, these results serve to highlight the potential usage of these simulations and inspire
future work on and with the modelling system.

### 3.1 Plausibility considerations

Figure 2 shows snow depth distributions observed at three contrasting forest plots in Laret during a spring survey on 17 April
2019, including a plot within-stand (upper row), a north-exposed forest edge (middle row) and a south-exposed forest edge
(lower row). These plots cover a broad range of canopy densities and insolation regimes, examples of which are show by the
hemispherical images displayed next to each plot's snow depth data. Consequently, within-plot snow depth variability is
large, and snow distribution patterns differ markedly between plots. Observations (right column) are compared to
simulations obtained with both FSMCRO (left) and FSM2 (middle), following the analysis presented in Mazzotti et al.
(2020b), see Figure 9 therein. Overall, the snow depth patterns resulting from the FSMCRO simulations are consistent with
the observations and comparable to the results obtained with FSM2, although FSMCRO appears to simulate slightly larger
snow depths than FSM2. Additional validation including error metrics and comparisons for a mid-winter survey at the same
sites as well as for a spring survey at Sodankylä are provided in the Supplementary Material S1, confirming this conclusion.
While a more detailed model validation is beyond the scope of this study, an adequate reproduction of observed snow depth
patterns is a prerequisite for a meaningful subsequent analysis of snowpack vertical properties. In this regard, Figure 2 attests
satisfactory performance of FSMCRO. Spatially distributed datasets that would allow to validate FSMCRO simulations at
the scale of individual snowpack layers do not exist at our sites to date. While FMI's long-term snow monitoring program
includes weekly snow pits in the forest (Leppänen et al., 2016), the changing and not exactly recorded locations of these pits
over the winter hamper the suitability of this dataset for use as validation data for our study's purpose.



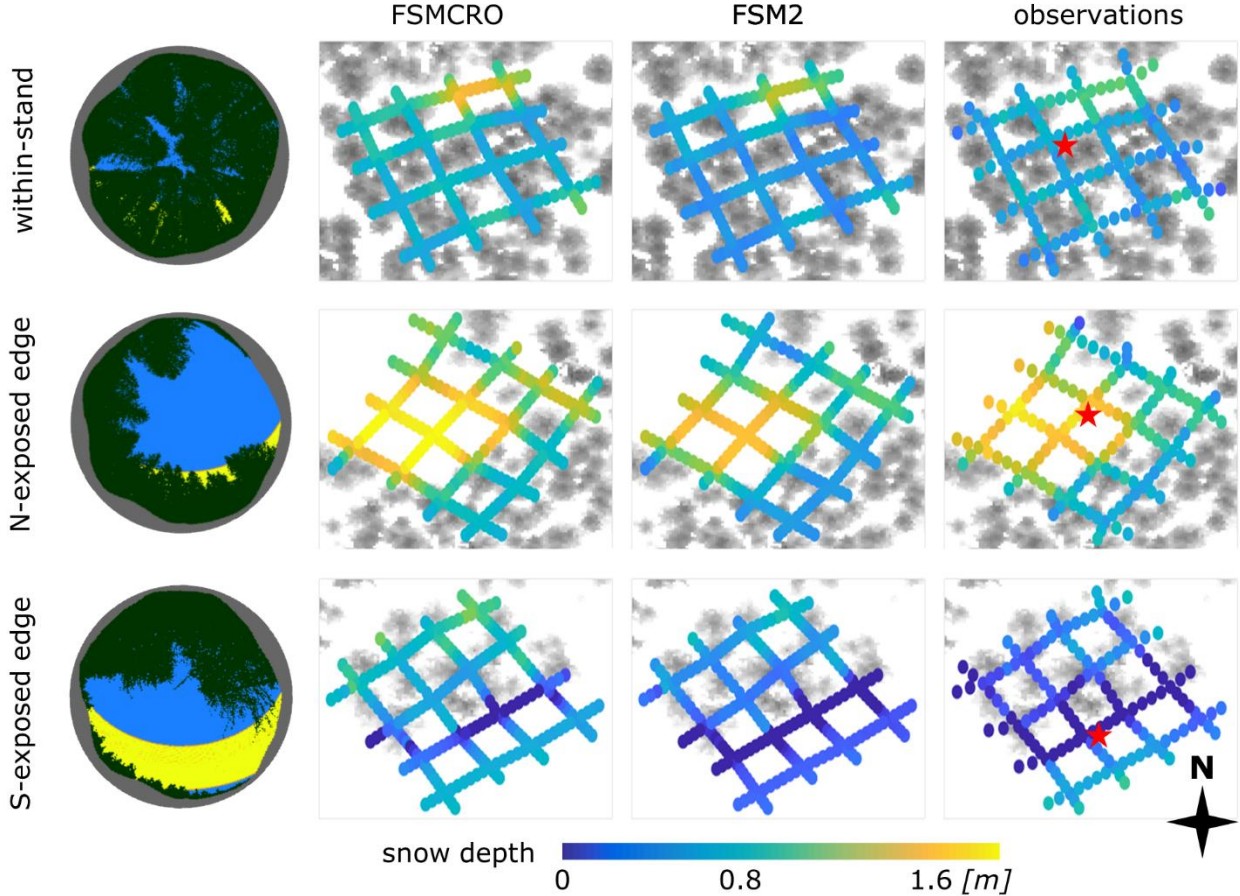

**Figure 2: Snow distribution observed on 17 April 2019 at three 50m x 50m forest plots in Laret and simulated with FSMCRO and FSM2, data from (Mazzotti et al., 2020b). Hemispherical images taken at the position of the red star are shown for each site, including canopy structure (grey/black), terrain (gray) and the solar tracks between 1 October and 17 April (yellow). These images**
**exemplify the characteristics of the contrasting within-stand, north-, and south-exposed canopy edges locations.**

## 3.2 Seasonal evolution of snowpack profiles at gap vs. under-canopy locations

An overview of snow properties accessible through FSMCRO simulations is given in Figure 3, contrasting closed-canopy and gap locations (upper vs. lower rows) in Sodankylä (blue frame) and Laret (green), respectively. The evolution of snow profiles is shown for SSA, snow grain type, temperature, and density. This is the standard way of visualizing layer-scale

properties, and mainly serves a qualitative comparison of different locations for the entire winter; nevertheless, some impacts of the presence of canopy on both accumulation and ablation processes are visible at both sites. The most prominent difference is the snow depth variability created by differences in interception during accumulation, and differences in melt rates, resulting in similar melt-out dates at under-canopy and gap locations despite more snow accumulating in the canopy gaps. A closer look at the snow stratigraphy reveals specific snowpack features that are present under-canopy but not in



gaps, and vice versa. At Sodankylä, for instance, the formation of surface melt forms/crusts (red) happens ca. 10 days earlier under-canopy than in the canopy gap, but the snowpack in the canopy gap features a much thicker bottom layer consisting of melt forms throughout winter.  At Laret, such examples include a layer of depth hoar + faceted crystals (light blue) formed close to the surface in January that persists longer in the canopy gap than under the canopy. Later in the season (mid-February), a surface melt crust (clearly visible yellow layer in the density profile) develops in the canopy gap but not under

closed canopy. Beyond the within-site comparison, it should further be noted that FSMCRO simulates substantially different stratigraphy in the two climates, as expected (Strum and Liston, 2021).

**3.3. Horizontal variability of snowpack stratigraphy along a forest discontinuity at different points in time**

Results presented in section 3.2 (Figure 3) provide evidence that at any given point in time, the spatial variability in canopy may translate to variability in layer-scale snowpack properties. To explore this more systematically, here we take a closer

look at the spatial variability in snow stratigraphy along a discontinuous forest transect, focusing our analysis on the examples of grain type and SSA. While still considering the vertical dimension, we now hence also fully resolve the horizontal dimension, which allows us to capture the complex influence of canopy edges as well. Replicating the concept of a long snow trench, Figure 4 presents temporal snapshots of snowpack stratigraphy, as characterized by grain type, along a ~100-m transect in Sodankylä over the course of the season. The transect crosses a large forest gap and thus encompasses

(from left to right) a north-exposed canopy edge, the canopy gap, a south-exposed canopy edge, and closed canopy (see dashed line in the canopy height model in Figure 1). Canopy structure is visualized at the top of the figure, quantitatively by local canopy cover fraction (Fveg) and conceptually (note the relative positions of the sun to the trees). The five selected timesteps cover a period of three months (mid-February to mid-May), approximately centered around peak of winter (end of March) and thus include scenes during both accumulation and ablation periods. They were chosen to capture typical

situations within the snow season: 1) melt crust formation in the forest following a snowfall (17 February); 2) subsequent burial of the crust by a snowfall event and its survival within the snowpack (12 March); 3) the onset of snowpack ripening following peak SWE (3 April); 4) the vertical progression of the melting front and associated disappearance of dry snow grain types (19 April); and 5) the onset of melt-out on the transect (10 May). The same temporal snapshots but showing SSA as an example of a quantitative snowpack property, are provided in Figure 5.



**Figure 3: Simulated snow profiles at contrasting within-forest locations at Sodankylä (blue frame) and Laret (green frame) including points under-canopy (upper rows) and in forest gaps (lower rows), see icons right of the plots. Profiles are shown for snow temperature (1st column), snow density (2nd), snow specific surface area (3rd) and snow grain type (4th). Grain type abbreviations correspond to the international classification for seasonal snow on the ground (Fierz et al., 2009) and are reported in Appendix A3.**



Several noteworthy features are revealed by these plots. Differences between closed canopy and canopy gaps mentioned in section 3.2 are confirmed and especially pronounced during accumulation: The surface melt crust visible on 17 February is consistently present at under-canopy locations but absent in the canopy gap, and likely results from drip unloading of intercepted snow at relatively warm air temperatures. Snow depth variability corresponding to the distribution of Fveg

develops during accumulation due to interception-related processes and remains the prominent pattern until around peak of winter. However, later in the season, different variability patterns along the transect evolve due to the variability in pathways of snow metamorphism, which arises from spatial differences in snow energy balance. This becomes especially visible in spring when the snowpack starts to ripen (3 to 19 April). Melt forms appear much earlier at the south-exposed edge of the gap than at the north-exposed edge. We also find that ripening is completed at the sun-exposed edge first, while other

locations still feature dry snow grain types (19 April). Overall, these results highlight that gradients in snow properties caused by heterogeneous canopy are highly dynamic in time. They further suggest that vertical snowpack heterogeneity generally outweighs horizontal variability during accumulation, but that the opposite may become the case during ablation, when vertical stratigraphy at individual locations becomes more homogeneous while horizontal differences between locations become more pronounced. Spatiotemporal patterns of SSA in Figure 5 underpin this tendency and show that the

variability suggested by the discrete snow grain type classification is associated with a remarkable variability of this continuous snow microstructure descriptor. The discrete grain type representation hence does not overestimate the variability of snow stratigraphy. For the sake of completeness, the same figures for snow temperatures and density are included in the Supplementary Material S2. They reveal that the spatio-temporal variability along the transect is considerable for all four snow state variables.

Ensemble simulations provide a means to assess if snowpack properties simulated at locations with specific canopy features are robust. Consequently, considering results from the full ensemble at locations that feature contrasting canopy structure reveals whether snowpack variability induced by heterogeneous canopy is greater than snow model uncertainty. Examples of such comparisons are shown in Figure 6 and include two of the timesteps shown in Figures 4 and 5 (accumulation vs. ablation). On 17 February, profiles simulated with the full ensemble are shown for a characteristic location within the canopy

gap and below closed canopy. All ensemble members exhibit a surface melt crust and only a thin bottom layer with melt form at the location in closed canopy. A surface crust consistently lacks at the canopy gap location across the ensemble, whereas a thick bottom layer with melt forms is present in all members. In contrast, snow depth variability between ensemble members at each location is in the same order of magnitude as differences between the two locations.







**Figure 4: Snow stratigraphy along a discontinuous forest transect at Sodankylä visualized in terms of snow grain type for five different dates covering the three-month period between mid-February and mid-May. Canopy structure along the transect is visualized in terms of local canopy cover fraction, Fveg.**





**Figure 5: Snow specific surface area at the layer scale along a discontinuous forest transect at Sodankylä for five different dates covering the three-month period between mid-February and mid-May. Canopy structure along the transect is visualized in terms of local canopy cover fraction, Fveg.**






For 19 April, results from the full ensemble are shown at two example locations at north- and south-exposed canopy edge, respectively. Differences between stratigraphies are much more marked than for the 17 February example. Most ensemble
members simulate a fully ripened snowpack at the south-exposed edge, but none do at the north-exposed one, where dry snow grain types prevail in the bulk of the snowpack across the ensemble at this locationn. Snow depth variability between ensemble members is pronounced at both locations, yet snow depths at the north-exposed edge are systematically larger than at the south-exposed edge. Some simulated snowpack properties appear to be less robust than others: at the north-exposed edge, the presence of melt forms at the surface is consistently simulated by all ensemble members, while depth hoar in the
lower part of the snowpack appears in most but not all ensemble members. While this implies that this snowpack feature is associated with greater uncertainty, discrepancies between the snowpack stratigraphy predicted by different ensemble members are much smaller than differences between stratigraphy simulated at the two contrasting locations (by any ensemble member). This finding provides strong evidence of the substantial impact of canopy structural heterogeneity on modelled snow stratigraphy, suggesting that the resulting variability by far exceeds model uncertainty.

**3.4. Spatial patterns and fractional partitioning of snow stratigraphy from fully distributed simulations**

While sections 3.2 and 3.3. put a strong focus on vertical snow profiles, here we show how fully distributed simulations can be used to assess the spatial patterns of specific snowpack properties and their relationship to canopy structure variability. As an example, Figure 7 shows maps of snow grain type for the surface layer simulated on the 400m x 250m domain in Laret at two points in time. The contribution of each grain type to the partitioning of the full domain is visualized with pie charts
corresponding to each time step. These charts reveal a highly heterogeneous snow surface at both dates, which reflects a strongly variable surface energy balance and subsequent melt and metamorphism. On 13 February, precipitation particles persist over three quarters of the domain, but melt forms prevail in the remaining quarter. Four days later, melt forms prevail over a large part of the domain but dendritic forms and faceted crystals persist at few locations. Comparing these maps to the canopy height map in Figure 1 evidences the links between snow grain type distribution patterns and canopy structure. For
instance, early appearance of melt forms along south-exposed forest edges and persistence of dry snow grain types along north-exposed forest edges imply that shading of the snow surface exerts a major control on metamorphism.







**Figure 6: Ensemble simulations of snow stratigraphy in terms of grain type at two contrasting locations within the forest transect from Figure 4, for the first and third timesteps shown therein. The upper scene contrasts points located in the canopy gap and in closed canopy during accumulation, while the lower scene compares north- and south-exposed canopy edges during ablation.**



As suggested by Mazzotti et al. (2021b) and Broxton et al. (2021), hyper-resolution forest snow simulations can be used to assess the explicit sub-grid variability that cannot be captured with spatially aggregated, coarser-resolution simulations.

Here, we contrast the distribution of surface grain type between the deterministic spatially explicit FSMCRO simulation, and the ensemble run obtained for one point that features canopy properties representing an average over the area. This allows us to assess whether unresolved variability in coarse-resolution simulations induced by forest structure is considerable compared to the uncertainty in snowpack simulations captured by the ensemble. Figure 8 shows the evolution of snow surface grain type partitioning over the domain from Figure 7 during February 2019, with data sampled at 6-hourly intervals.

The time series evidences that canopy-mediated processes induce strong spatial heterogeneity during some periods, but not in others. Strong spatial variability, evidenced by the presence of many different grain types during a specific time step, notably occurs in phases following snowfall events when metamorphism occurs at variable rates across the domain. During these periods, the spatially averaged ensemble simulations feature a much more marked transition from precipitation particles to melt forms, while the deterministic fully-distributed simulation reveals a phase during which melt forms and dry

snow types co-exist at the surface. The ensemble thus does not capture variable metamorphism rates that are tightly linked to specific canopy structures. Overall, the hyper-resolution simulation reveals more diverse snow surface conditions and smoother transitions between dry and wet snow regimes.

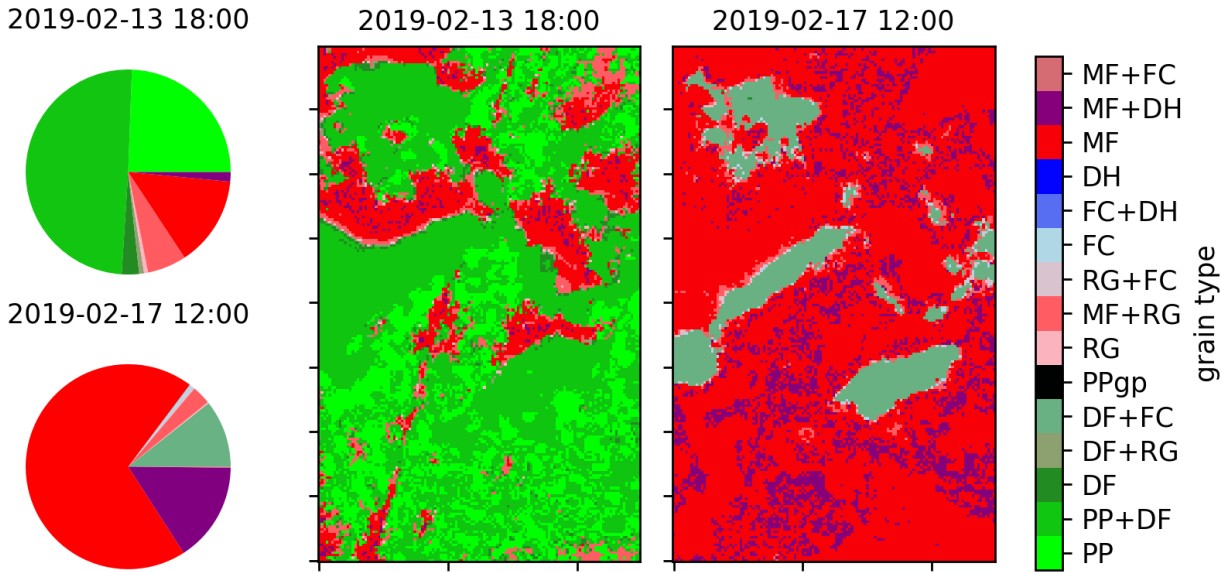

**Figure 7: Pie charts and maps visualizing the fractional partitioning of snow surface properties in terms of snow grain type over**
**the 400m x 250m domain in Laret during two timesteps in February 2019.**



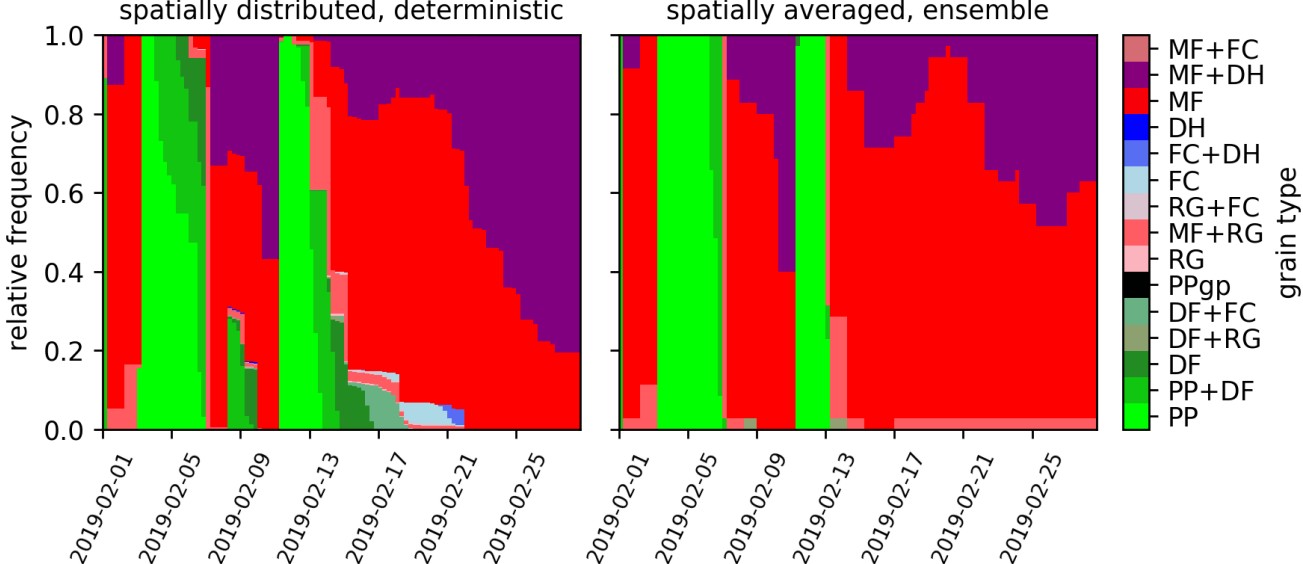

**Figure 8: Temporal evolution of grain type partitioning over the 400m x 250m domain shown in Figure 7 during February 2019 at a temporal resolution of 6 hours, as derived from the deterministic, spatially distributed simulation at 2m resolution (left) and from the ensemble members of the point simulation representing spatially aggregated canopy properties.**

## 4 Discussion

### 4.1 FSMCRO – a new hyper-resolution forest snow modelling tool

For the first time, this study presented snow stratigraphy simulations obtained with a microstructure-resolving snow physics model that sees the explicit impact of forest canopy structure on sub-canopy micrometeorological conditions at the meter scale. To our knowledge, this type of forest snow simulations at a comparable spatial resolution had previously only been attempted by Perrot et al. (2014). In their study, however, the impact of fine-scale canopy structure on meteorological variables other than snowfall (via interception and unloading) was disregarded, neglecting accurate representation of sub-canopy irradiance patterns. The establishment of the FSMCRO model framework, which overcomes this limitation and made our simulations possible, hence constitutes a major contribution. Its development was a logical next step following recent progress in hyper-resolution forest snow modeling; however, an attempt to fully couple Crocus and the FSM2 canopy shed light on numerical pitfalls that are likely not uncommon but poorly documented in literature. For instance, Cristea et al. (2022) showed that choices of snow layer number and thicknesses can considerably impact snow simulations; such potential model artifacts are difficult to identify and can be problematic, especially for model application studies.

The goal of the relatively simple approach chosen for FSMCRO was to develop a modelling system that avoids numerical issues and facilitates its integration in other model frameworks, in the interest of transferability and easy usage in future applications. In this context, using meteorological transfer functions instead of a fully coupled canopy implementation




entails some key advantages. Firstly, it allows for the same type of boundary conditions at open and forested locations, which ensures that all options contained in the ensemble modeling framework ESCROC are usable. This is not (yet) the case with a MEB-like canopy implementation, as outlined by Nousu et al. (2023). Moreover, although most transfer functions

used here are relatively simple, each variable can be modified separately where more sophisticated, locally calibrated parametrizations are available, such as the sub-canopy longwave radiation model by Webster et al. (2017), or alternative interception and unloading functions as considered in Lumbrazo et al. (2022). Lastly, the concept is not specific to Crocus and should be easily transferable to other complex snow models that have so far mainly been applied at open sites, e.g., SNOWPACK (Bartelt and Lehning, 2002). It should be noted that meteorological transfer functions have been used to apply

snow models without canopy implementations to forested sites before (Gelfan et al., 2004; Perrot et al., 2014; Bonner et al., 2022a).

As main drawback, meteorological transfer functions do not yield canopy states and are therefore not suitable for model frameworks that require these to be tracked (i.e., integrated land surface and atmospheric model systems). Yet, for most applications the FSMCRO codebase provides a convenient framework to test coupled canopy representations in the future.

Such efforts should opt for a model structure that enables (or at least approximates) tight coupling of the canopy and snow model components by solving both the canopy and the snowpack energy budget in the same equation system. Corresponding numerical approaches have been proposed, e.g., by Boone et al., (2017) and Clark et al. (2015).

## 4.2 New insights on the impact of canopy structure on snow stratigraphy

The proposed FSMCRO system allowed for snow simulations that capture how canopy structure variability translates to

variability in snow stratigraphy and associated microstructural properties. Spatially distributed ensemble simulations contain an enormous amount of information that is not accessible through simpler models or observations, including the horizontal, vertical, and temporal variability of multiple snowpack properties and their associated uncertainties. Results presented in section 3 showcase some ideas on how to analyze this multidimensional data and yielded first insights on the impact of fine-scale canopy characteristics on layer-scale snow properties.

Our simulations indicate that canopy structure can have a considerable impact on snow stratigraphy. We identified two main canopy-mediated mechanisms that create variability in our simulations, with different spatio-temporal patterns. First, drip unloading of intercepted snow at above-freezing temperatures that occurs at under-canopy locations but is absent in gaps created layers that are only found under closed canopy, typically consisting of melt forms. Second, variations in energy input to the snow surface, which are strongly linked to insolation patterns, entail spatial variability in metamorphism, yielding

heterogeneous patterns of dry grain types and melt forms. This effect requires sufficiently high solar radiation input to be relevant and is therefore especially notable after peak of winter. The first mechanism is in line with findings from Bonner et al. (2022b), who observed a clear signature of canopy unloading events in snow density profiles under canopy using SUMMA model point simulations. The second mechanism has, to our knowledge, not been captured by any simulation prior to this study.





An interesting consequence of these two mechanisms is that different patterns of spatial variability arise at different times in the snow season. Early in the winter, horizontal variability is rather small, and contrasts occur mainly between closed-canopy and canopy gap environments. As shortwave irradiance increases, insolation-driven variability in snow metamorphism and snowpack ripening can cause stratigraphy to be highly heterogeneous over short distances. Canopy edge environments with opposite orientation hereby contribute to enhancing this spatial heterogeneity further. Near the snow surface, spatial 480 variability is strongly governed by snowfall events: while every snowfall creates homogeneous snow surface conditions for a short period, the subsequent metamorphism create strongly heterogeneous patterns that persist until surface melt is attained everywhere (or a new snowfall event occurs).

We further demonstrated the use of ensemble simulations and hyper-resolution simulations in tandem to assess whether stratigraphy differences induced by canopy structure are robust relative to model uncertainties. Examples shown in section 485 3.3 include distinct features at specific locations, and differences between locations that are captured by a majority of the ensemble members. In such cases, we could confirm that simulated snowpack variability induced by canopy structure is indeed larger than estimated snow model uncertainty. A consequence of this finding is that the spread of ESCROC simulations cannot be considered as a proxy of sub-grid variability for coarse-resolution simulations over heterogeneous forest landscapes, as shown in section 3.4. This has important implications for future applications of ESCROC in forests. For 490 instance, the fact that snowpack properties not captured by ESCROC are associated with specific canopy structure features would entail systematic, canopy-dependent model errors. A better understanding of where and when such errors occur would increase the utility of microstructure-resolving forest snow cover simulations, e.g., in the context of snow remote sensing applications.

While available observations did not allow to evaluate our simulations in more detail, our findings are generally consistent 495 with the few observational studies that have addressed snow microstructural properties, stratigraphy, and its spatial variability within the forest. Comparing snow pits in canopy gaps and closed canopy, Bouchard et al. (2022) identified layers specific to closed-canopy locations that were absent in forest gaps. Optical grain size measurements in trenches along tree boles presented by Molotch et al. (2016) revealed distinct differences in snow grain size at small vs. large distance from tree trunks, and faster metamorphism at south- vs. north- exposed sides of trees. Based on SnowMicroPen measurements along 500 transects, Teich et al. (2019) found greater spatial heterogeneity in snow stratigraphy in within-stand transects than in open terrain. Results from our simulations are in line with all these experimental findings.

## 4.3 Paving the way for future forest snow model applications

The amount of information accessible through FSMCRO simulations creates plenty of opportunities for novel model applications. Besides avalanche research, several disciplines could benefit from snow microstructure information in forests. 505 Snowpack properties such as surface density and presence of ice layers are key features impacting wildlife ecology, affecting animal movement, pray-predator interactions, winter habitats, and foraging success (Sullender et al., 2023; Cosgrove et al., 2021). Gas transfer through the snowpack is also impacted by microstructural properties, with implications for wintertime





greenhouse gas exchange in boreal environments (Pirk et al., 2016; Graham and Risk, 2018). Finally, knowledge of snow
microstructure and/or layer-scale properties is key for the interpretation of certain remote sensing retrievals, e.g. microwave

remote sensing and ground-penetrating radar (Webb et al., 2018; Picard et al., 2018). In all these contexts, application of a
model like FSMCRO holds great potential to advance research and facilitate studies in forested environments.

Ultimately, such detailed simulations also increase our understanding of the modelled system. The scale gap between the
true spatial variability of snow-vegetation interactions and typical resolutions of hydrological and land surface models has
long been a major challenge for modelling forest snow due to the difficulty of linking point-scale observations to simulations

(Clark et al., 2011; Fassnacht, 2021). The framework proposed here constitutes an additional step in overcoming this gap.
Linking specific snowpack features to specific locations in heterogeneous canopy structure based on hyper-resolution
simulations could, for instance, motivate strategies for the representation of sub-grid variability in coarse-resolution models.
Alternatively, FSMCRO simulations could be used as benchmark for ESCROC applications to forests at coarser resolutions.
For example, our results suggest that canopy heterogeneity, e.g., in terms of LAI perturbations or canopy parameter sets

specific to characteristic forest locations, could be added as additional option to increase ensemble spread. This would allow
to better capture unresolved spatial variability arising from heterogeneous canopy structure in ensemble simulations, or to
estimate model uncertainty in forest simulations for which canopy descriptors are poorly constrained. Generally, the recent
strong push towards replacing computational bottlenecks by emulators trained on detailed, physics-based high-resolution
models in geoscientific and cryospheric research (Jouvet, 2023; Uchôa da Silva et al., 2022) underlines the current need for

further development and improvement of such detailed, physics-based modelling approaches.

## 4.4 Identifying priorities for follow-up research

Because the purpose of this study was to provide a proof of concept, we used FSM2 and ESCROC out of the box rather than
putting effort into finetuning our simulations to match available validation data. Yet, our results already revealed some
potential for further model enhancement, as well as a current need for additional validation data and improved approaches to

analyze the multidimensional simulation results which would support such model development.

Results from section 3.1 suggest that FSMCRO slightly overestimates snow depth on average and slightly underestimates its
variability compared to FSM2. A possible reason is that parametrizations of snowpack properties in Crocus do not take
forest into account. In FSM2, in contrast, roughness length and albedo parametrizations were modified as a function of
canopy density to mimic the effect of litterfall on snow surface albedo and the impact of a heterogeneous surface on

roughness length (Mazzotti et al. 2020c). While adaptations to the Crocus model were beyond the scope of this study,
exploring whether its parametrizations can be adapted to better represent forest conditions should be addressed in future
studies. Beyond adjustments to the default albedo parametrizations, Crocus' recent developments to incorporate impurities
(Tuzet et al., 2017) provide an interesting avenue that could be extended to litterfall. Moreover, a microstructure-resolving
model like Crocus is expected to be more sensitive to the phase and properties of unloading snow than intermediate-

complexity snow models, but this aspect has so far not been investigated and would deserve further attention. Further,



Crocus is also more sensitive to initial soil conditions (Lafaysse et al., 2017); here, we focused on the snow season and provided homogeneous initial conditions to ensure compatibility with Mazzotti et al. (2020b,c), but differential shading might also imply differential initial conditions at the soil interface. Interdisciplinary efforts involving datasets acquired by a growing community of forest microclimate researchers (Lembrechts et al., 2020) would, apart from extended model spin-up,

allow to refine initial soil conditions for future FSMCRO applications at hyper-resolutions.

Any further model development effort would obviously benefit from more evaluation data. However, even where snow stratigraphy data is available (e.g., Bonner et al., 2022), comparison with model results is not straightforward due to unavoidable discrepancies in total snow depth and often limited vertical resolution of traditional snow pit measurements (e.g., Leppänen et al., 2016), requiring the application of layer-matching algorithms (Hagenmuller and Pilloix, 2016) that

still face unsolved methodological challenges (Viallon-Galinier et al., 2020). For validating FSMCRO simulations, sampling multiple locations within the forest would be indispensable, and our findings could inform suitable sampling strategies to cover contrasting snow stratigraphies. Yet, besides validation data, new approaches to quantitatively analyze the multidimensional information accessible through the simulations also need to be developed. Existing layer matching algorithms target the comparison of observations and simulations at the same locations, but fewer approaches exist to

quantitatively compare profiles from different locations (Herla et al., 2023).

## 5.    Conclusion

For the first time, this study merged the capabilities of a tree-scale forest snow and a detailed snow physics model to enable layer- and microstructure-resolving forest snow simulations at 2-m spatial resolution, including uncertainty estimates with an ensemble. Application to sub-alpine and boreal forests, intended primarily as a proof of concept, evidenced a strong

influence of tree-scale canopy properties on the evolution of snow stratigraphy, with marked spatiotemporal patterns arising 1) as a result of interception and subsequent drip unloading of canopy snow, which occurs in closed canopy but not in canopy gaps, and 2) due to insolation that drives spatial differences in snow energy balance and subsequently also in metamorphism and melt, for instance between canopy edges of opposite orientation. Our results underline the potential of FSMCRO simulations as a tool for scientific applications. The multidimensional information provided by these simulations,

covering different state variables, their vertical and horizontal variability, temporal evolution, and uncertainty assessment, is unprecedented and cannot be obtained with observational methods or simpler models. It thus constitutes a valuable complement to existing tools and can contribute to advancing our understanding of forest snowpack dynamics, for the benefit of several research disciplines. Besides inspiring new applications, we hope that this work will also motivate further data acquisition towards continued model validation and enhancement.



**Appendix**

**A1. Adjustments to above-canopy meteorological forcings to represent below-canopy conditions**

FSMCRO requires meteorological inputs of snow- and rainfall rates, direct and diffuse shortwave as well as longwave radiation, air temperature and relative humidity, surface pressure, and wind velocity. When driving a forest snow model with station data, meteorological forcing is usually from either above the canopy or a nearby open site. In the latter case, it is

commonly assumed that measurements from the open site correspond to above-canopy conditions. The transfer functions applied with FSMCRO then aim at adjusting above-canopy meteorology to below-canopy conditions.

For FSMCRO, the idea was to keep the adjustments as simple and close to FSM2 as possible, while at the same time accounting for the specific impact of different characteristics of the canopy on different meteorological variables by means of process-specific canopy structure metrics as implemented in FSM2. Consequently, the below equations heavily rely on

Mazzotti et al. (2020a,b) and Appendices therein.

The below-canopy reference level was chosen to be 2 m, which means that the ESCROC snow scheme(s) 'see' the meteorological forcing as if it was measured at a given height above ground. In the following, the subscript 'a' denotes above-canopy (or atmospheric) meteorological conditions, while the subscript 'c' refers to the adjusted below-canopy variables.

Like in FSM2, above-canopy direct and diffuse shortwave radiation components $SW_{a,b}$ and $SW_{a,d}$ are scaled by the respective transmissivities ($\tau_b$ and $\tau_d$) and summed to obtain total sub-canopy incoming shortwave radiation:

$$SW_c = \tau_b SW_{a,b} + \tau_d SW_{a,d}$$

Where $\tau_b$ is provided as time-varying input to the model, while $\tau_d$ is static and corresponds to hemispherical sky-view fraction.

Atmospheric longwave radiation is enhanced by thermal radiation emitted by the canopy, where the weighting of atmospheric and canopy components is based on sky-view fraction (i.e. $\tau_d$), as in FSM2. However, canopy temperature is assumed to equal air temperature ($T_a$, provided as forcing), unlike in FSM2 (where canopy temperature is a state variable).

$$LW_c = \tau_d LW_a + (1 - \tau_d)\sigma T_a^4$$

Below-canopy wind velocity is obtained using the wind profiles implemented in FSM2, where the wind profile $U_{sc}$ at any

location with a given stand-scale canopy cover fraction $f_{vs}$ (canopy structure input of FSM2, computed equivalently to canopy top height $h$) is as weighted average of the open-site logarithmic profile ($U_{opn}$) and wind profile corresponding to dense canopy ($U_{dc}$):

$$U_{sc}(z) = f_{vs}^{0.5} U_{dc}(z) + (1 - f_{vs}^{0.5}) U_{opn}(z)$$

A composite log-exp wind profile is assumed in dense canopy, where decay is logarithmic above the canopy, exponential

from the canopy top $h$ to a reference level $z_{sub}$ below the canopy (in FSM2, $z_{sub}$ = 2m), and logarithmic between $z_{sub}$ and the ground



$$U_{dc}(z) = \begin{cases} U_a \, ln \frac{z-d}{z_{0v}} \left[ ln \frac{z_U-d}{z_{0v}} \right]^{-1} & z \geq h \\ U_{dc}(h) e^{\eta(z/h-1)} & z_{sub} \leq z < h \\ U_{dc}(z_{sub}) \, ln \frac{z}{z_{0g}} \left[ ln \frac{z_{sub}}{z_{0g}} \right]^{-1} & z < z_{sub} \end{cases}$$

where $h$ denotes stand-scale canopy height (i.e. computed over a 50m radius around a point, canopy structure input of FSM2), $d = 0.67 \, h$ is zero-plane displacement, $z_{0v} = 0.1 \, h$ is vegetation roughness length, $\eta = 2.5$ is a wind decay factor and $z_{0g}$ is the ground roughness length. Wind speed at the below canopy reference height is thus obtained as $U_{sc}(z)$ with $z = 2$ m. Snow- and rainfall rates are modified to account for interception of snow in the canopy and its subsequent unloading and sublimation. Interception and unloading parameterizations, based on Hedstrom and Pomeroy (1998), are identical to those implemented in FSM2, including local canopy cover fraction $f_{vl}$ and Leaf Area Index as canopy descriptors to yield realistic spatial variability. Over each time step $\delta t$, the increase in intercepted snow mass $\delta S_v$ is:

$$\delta S_v = (S_{\max} - S_v) \left[ 1 - \exp\left( -\frac{f_{vl} S_f \delta t}{S_{\max}} \right) \right]$$

where $S_{\max} = 4.4 \text{LAI}$ is the maximum canopy snow holding capacity. Snow unloads from the canopy at rate $\tau_u^{-1} S_v$ with different values of the time constant $\tau_u$ for cold and melting snow. Unloading snow is added to snowfall if air temperature is below freezing and as rainfall otherwise.

Interception of rainfall is not accounted for. All other meteorological variables (air temperature, surface pressure, and relative humidity) are left unchanged from open site conditions. The conversion of relative to specific humidity, which is needed by Crocus, is implemented in the model.

**A2. Crocus options used in the deterministic and ensemble simulations**

Table A1 lists the combinations of Crocus options used for the deterministic run (operational configuration, Vernay et al., 2022) as well as for the ensemble members applied in this study. Abbreviations used to denote the different Crocus options follow Lafaysse et al. (2017).

| | **Crocus options** | | | | | | | |
|---|---|---|---|---|---|---|---|---|
| | Snowfall density | Metamor phism | Solar radiation | Turbulent surface fluxes | Thermal conduc-tivity | Liquid water retention | Compac-tion | Surface heat capacity |
| **Deterministic run** | V12 | C13 | B60 | RI1 | Y81 | B92 | B92 | CV30000 |
| **Ensemble (member)** | | | | | | | | |
| 1 | V12 | C13 | B60 | RI1 | Y81 | SPK | B92 | CV30000 |
| 2 | V12 | C13 | B60 | RI1 | I02 | B92 | S14 | CV30000 |
| 3 | V12 | C13 | B60 | RI2 | Y81 | B92 | S14 | CV30000 |



| 4 | V12 | C13 | B10 | RIL | Y81 | B92 | B92 | CV30000 |
|---|-----|-----|-----|-----|-----|-----|-----|---------|
| 5 | V12 | C13 | B60 | RI1 | I02 | SPK | T11 | CV50000 |
| 6 | V12 | C13 | B60 | RI2 | I02 | SPK | S14 | CV50000 |
| 7 | V12 | C13 | B10 | RI1 | I02 | SPK | B92 | CV30000 |
| 8 | V12 | B21 | B60 | RIL | Y81 | B92 | S14 | CV30000 |
| 9 | V12 | B21 | B60 | RIL | Y81 | SPK | S14 | CV50000 |
| 10 | V12 | B21 | B60 | RIL | I02 | B92 | T11 | CV50000 |
| 11 | V12 | B21 | B60 | RI1 | I02 | B92 | S14 | CV30000 |
| 12 | V12 | B21 | B10 | RI2 | I02 | SPK | B92 | CV30000 |
| 13 | V12 | B21 | B60 | RIL | Y81 | SPK | S14 | CV50000 |
| 14 | V12 | B21 | B60 | RIL | Y81 | SPK | S14 | CV30000 |
| 15 | V12 | B21 | B60 | M98 | Y81 | SPK | B92 | CV30000 |
| 16 | V12 | B21 | B10 | RIL | I02 | SPK | B92 | CV30000 |
| 17 | S02 | C13 | B60 | M98 | Y81 | B92 | S14 | CV30000 |
| 18 | S02 | C13 | B10 | RI1 | I02 | B92 | B92 | CV30000 |
| 19 | S02 | B21 | B60 | RIL | Y81 | B92 | S14 | CV50000 |
| 20 | S02 | B21 | B60 | M98 | I02 | B92 | B92 | CV30000 |
| 21 | S02 | B21 | B60 | M98 | I02 | SPK | B92 | CV30000 |
| 22 | S02 | B21 | B60 | RI1 | Y81 | SPK | B92 | CV30000 |
| 23 | S02 | B21 | B10 | RIL | I02 | SPK | B92 | CV30000 |
| 24 | S02 | B21 | B10 | RI1 | I02 | SPK | B92 | CV30000 |
| 25 | S02 | B21 | B60 | RIL | I02 | B92 | B92 | CV50000 |
| 26 | S02 | B21 | B60 | RIL | I02 | SPK | S14 | CV50000 |
| 27 | S02 | B21 | B60 | RI1 | I02 | SPK | B92 | CV30000 |
| 28 | S02 | B21 | B60 | RIL | I02 | SPK | S14 | CV50000 |
| 29 | A76 | B21 | B60 | M98 | I02 | B02 | S14 | CV30000 |
| 30 | A76 | B21 | B60 | M98 | I02 | SPK | B92 | CV30000 |
| 31 | A76 | B21 | B10 | RIL | I02 | B92 | B92 | CV30000 |
| 32 | A76 | B21 | B10 | RIL | Y81 | B92 | B92 | CV30000 |
| 33 | A76 | B21 | B10 | RI2 | I02 | B92 | B92 | CV30000 |
| 34 | A76 | B21 | B60 | RIL | Y81 | SPK | S14 | CV50000 |
| 35 | A76 | B21 | B60 | RI1 | Y81 | SPK | B92 | CV30000 |

**Table A1: Crocus options used in the deterministic run as well as in each member of the ensemble used for this study.**

**A3. Snow grain type classification**

Abbreviations for the snow grain types used in Figures 4 and 6-8 are defined in Figure A1 and displayed alongside the corresponding categorical color codes, following Fierz et al. (2009). Colors in the first column correspond to categories consisting of each specific snow grain type only. Colors in the second column correspond to 'mixed' categories composed of multiple grain types and include all mixed categories in which the corresponding grain type occurs (e.g., the purple color being present in the rows of MF and DH indicate the category MF+DH). Only categories that contain melt forms are considered 'wet' snow types.



| | | | | MF | **M**elt **F**orms |
|---|---|---|---|---|---|
| | | | | DH | **D**epth **H**oar |
| | | | | FC | **F**aceted **C**rystals |
| | | | | RG | **R**ounded **G**rains |
| | | | | DF | **D**ecomposing / **F**ragmented precipitation particles |
| | | | | PP | **P**recipitation **P**articles |
| | | | | PPgp | **G**raupel **P**recipitation |

**Figure A1: Snow grain type names, abbreviations, and color codes**

*Code and data availability.* SURFEX is an open-source project (http://www.umr-cnrm.fr/surfex) but requires registration; instructions are available at https://opensource.umr-cnrm.fr/projects/snowtools_git/wiki/Procedure_for_new_users. The standalone Crocus version used in this work is available in git (tag: s2m_top_202305). The FSMCRO code is available on GitHub (https://github.com/GiuliaMazzotti/FSMCRO). Datasets are all published and referenced accordingly.


*Author contributions.* This study was designed by GM, ML and TJ. RN, ML and VV developed the standalone Crocus version. GM implemented the FSMCRO system, conducted the simulations, and analyzed the results, with support from JPN, ML, VV and TJ. GM wrote the paper, with contributions and feedback from all co-authors.

*Competing interests.* The authors declare that they have no conflict of interest.

*Acknowledgements.* GM was funded by a Postdoc.mobility grant from the Swiss National Science Foundation (Project: TP500PN_202741). JPN was funded by the Research Council of Finland (ArcI Profi 4) and the EU Horizon Europe Framework Programme for Research and Innovation (GreenFeedBack project; no. 101056921). Fieldwork in Sodankylä was
supported by InterACT (grants IME4Rad and UpForSnow). We thank Léo Viallon-Galinier and Mathieu Fructus for their substantial work on the snowtools visualization software and their support with it; Richard Essery, Kévin Fourteau, Julien Brondex, Aaron Boone, and Isabelle Gouttevin for insightful discussions on model coupling; Bertrand Cluzet for help with the ensemble simulations; Marie Dumont and Jessica Lundquist their scientific input and support of this project; and Antoine Corcket and Antoine Courteaud for advancing Crocus applications in the forest during their internships at the CEN. Canopy
structure and meteorological datasets used in this study would not exist without the valuable work of Clare Webster and Johanna Malle.



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
