# Peer review of "Exploring the potential of forest snow modelling at the tree and snowpack layer scale"

_EGUsphere, 2023_

## Author Comment (AC1)

**Exploring the potential of forest snow modelling at the tree- and snowpack layer scale – Response to reviewer # 1**

**Summary and recommendation**

Mazotti et al. develop and present a new physics-based, multi-layer, hyper-resolution snow model (FSMCRO) that can represent high spatial and vertical resolution snow properties including grain type, density, temperature, and other snow parameters. This was achieved through a one way coupling between the FSM2 canopy model and the ensemble Crocus model, with the added benefit that ensemble simulations provides a mean for assessing uncertainty. The paper focuses on introducing and demonstrating the model at two well studied snow sites (Finland and Switzerland), with only qualitative validation ("plausibility"). The model shows reasonable representation of snow depth patterns in Switzerland (focus in the main paper) but less so in Finland (supp. material). Overall, the model shows realistic spatial variations in key snow properties (grain size, SSA) and their evolution in time along a transect spanning a forest gap with variable radiation and interception dynamics. Through the use of the ensembles and spatial simulations, the study also finds that snowpack variability (due to canopy effects on snow processes) is more important than model uncertainty.

Overall, I find this paper potentially offers a significant advance in our ability to resolve very localized snow properties which will be of interest and use to research in snow-forest interactions, wildlife ecology, and possibly avalanche studies. I think the scientific and presentation are generally of high quality, though I offer some comments and suggestion for further improvement. My main concern is about the minimal validation effort and the apparent deficiencies in snow depth simulation at one of the sites (See #1 below), and therefore request the authors consider these before publication. I emphasize this paper should be published following attention to these comments.

We would like to thank the reviewer for the careful review, the overall positive assessment of our work, and the constructive comments. Please find our answers to each comment below in blue.

**MAIN COMMENTS**

1. While the paper does not present a detailed validation but rather a demonstration of the new model, it seems there is still an opportunity to provide additional analysis to understand the "plausibility" of the model and needs for future improvements. For instance, the paper references weekly snow pit data at the Finland site, but does not make use of them due to issues with geolocation. I would argue that the geolocation issue with the pits does not preclude such a comparison, as multiple location from the domain could be selected, along with the ensemble members in order to understand the range of possible snow profiles simulated by FSMCRO. I think that a comparison between the FSMCRO ensemble and the snow pit data (grain type, density, etc.) could still be informative, even if done on a qualitative basis given the recognized challenges in comparing multi-layer snow models to snow pits. This might help to identify the plausibility of the model as well as possible deficiencies and areas for future development in the model. At the same time, this may require attention to the prominent errors in FSMCRO snow depth that are apparent at the Finland site (Figure S2, where even normalized snow depths are quite different from observations). As noted by the authors: "an adequate reproduction of observed snow depth patterns is a prerequisite for a meaningful subsequent analysis of snowpack vertical properties" (L. 285-286). Comparing to the Finland snowpit data might be helpful for diagnosing possible reasons for the deficient snow depth representation (e.g., bulk snow density?).

Indeed, validation should not be a major component of this study, yet we understand the reviewer's point and have further investigated the two issues mentioned. Regarding the relative snow depth patterns in Sodankylä (Fig S2), we have identified two reasons that have contributed to the differences in simulation between FSMCRO and FSM2:

1) For both simulations we have used a precipitation undercatch correction that was determined based on FSM2 simulations at the open site. This is a standard procedure in case of FSM (e.g., Essery et al.

2017, https://doi.org/10.5194/gi-5-219-2016), but not for Crocus (e.g., Nousu et al., 2023, https://doi.org/10.5194/egusphere-2023-338). Since this work was intended as a first model demonstration, with thorough model validation and tuning to follow separately, we left this issue unattended. However, we repeated the simulations of FSMCRO without undercatch correction, which improves the match with observations (see Figure below, lower right panel).

2) The ablation rates of FSMCRO are somewhat lower compared to FSM2, again, owing to the lack of any specific model tuning of FSMCRO at this point. Consequently, snow distribution patterns that better match those observed are only attained a few days later (see Figure below, upper right panel). The below-canopy albedo or the subcanopy turbulent exchange parameters may require some tuning to fix this issue, as noted in the discussion.

[Figure]

Regarding a comparison of FSMCRO simulations with snow pit data, we believe this deserves full attention in a follow up paper, as shown by various examples of dedicated studies involving snow pit validations (Bouchard et al., 2024: https://doi.org/10.5194/tc-18-2783-2024; Calonne et al. 2020, https://doi.org/10.5194/tc-14-1829-2020; Leppäenen et al. 2017, https://doi.org/10.3189/2015JoG14J026 ). Yet, to address your comment we explored first test simulations. Note however that forest snow pits were only available since WY 2019, which does not correspond to the study period considered in the paper. We therefore had to perform simulations for additional years at a range of locations that appear to be qualitatively 'similar' to the area where the snow pits were located (which is outside of our study area). Comparison of these simulations (lines) with snow depths recorded in the snow pits (dots) is shown in the figure below.

[Figure]

First exemplary comparisons of FSMCRO simulations with snow pit observations are shown below. We compared simulated and observed density and temperature profiles (for WYs 2020 and 2022, corresponding to above average and average snow conditions), as well as ensemble simulations for individual survey days (see below). These analyses will be added and discussed in the Supplementary Material. A comparison of grain type however was beyond the scope of this additional analysis and shall be left for a dedicated follow-up study. As noted by Leppänen et al. (2016; https://doi.org/10.5194/gi-5-163-2016), this measurement is the most prone to observer-related bias. Moreover, the frequent use of subclasses and mixed grain types as well as the conversion to a quantitative metric would require substantial quality control and postprocessing.

[Figure]

[Figure]

2. Several figures in the paper are not readable for someone with a red-green vision deficiency. As such, those readers may not be able to distinguish (for instance) the different snow grain types (e.g., melt forms vs. precipitation particles). I recognize this is not the fault of the authors as they are following the conventions from the Fierz et al. (2009) international snow classification report. However, I would suggest the authors consider whether something can be done to help these readers (e.g., adding a small hatch pattern to the green colors).

Thank you for pointing this out, we recognize that our figures showing grain types are not adapted to red-green vision deficiency. However, this colormap is an international convention (see Fierz et al. (2009)) that required substantial work to be established and is now widely accepted in the community. We believe it is important to be consistent with this convention and will therefore not change the figures in the main article. If providing an alternative colormap is a required addition rather than an optional suggestion, we will add an adapted version of all relevant figures in the Supplementary Material, please let us know.

3. I recommend adding snow hardness and snow liquid water content (LWC) as new figures in the supplement (similar to Figures S3-S4), as the capability for mapping these variables spatially may be of high interest to other researchers. The paper references wildlife ecology, and for that the snow hardness is a relevant parameter. Likewise, snowmelt studies and microwave remote sensing (e.g., GPR) may benefit from a model that can resolve spatial variations in LWC.

We will add figures showing liquid water content and ram resistance (preliminary version see below) to the Supplementary Material and refer to it in the main text, thank you for the suggestion. The ram resistance is available as a diagnostic variable in Crocus and a commonly used proxy for snow hardness (see Fierz et al. 2009).

[Figure]

[Figure]

**Line Comments**

- L. 27: This should be "tools". Thank you for the catch, this will be corrected.

- L. 30-35: The opening sentence is rather long and cumbersome. I recommend breaking it into two or more sentences. The sentence will be broken down into five shorter sentences as follows: 'Seasonal snow takes many roles in the Earth's systems. As part of the land surface, it acts as reflective and insulating material, substantially influencing the Earth's energy budget (e.g., Thackeray and Fletcher, 2016; Colman, 2013; Sturm et al., 1997). Snow further constitutes an important seasonal storage of water, shaping the hydrograph of snow-dominated catchments (e.g., Barnhart et al., 2016; Bales et al., 2006; Viviroli and Weingartner, 2004). Snow is also a crucial ecosystem and habitat component in many regions, affecting animal movement, food accessibility, and soil thermodynamics and biogeochemistry (e.g., Boelman et al., 2019; Gilbert et al., 2017; Stark et al., 2020; Zhang et al., 2018). Lastly, snow avalanches are a common natural hazard in many mountain regions (Schweizer et al., 2003).'

- L. 215: Add "an" after "as". Will be added.

- L. 256: The sentence begins with awkward wording. Please rephrase. The sentence will be rephrased as follows: 'Merely, the snow metamorphism options based on Flanner and Zehnder (2006), which are not yet available in the standalone version of Crocus, were replaced by an improved metamorphism parametrization recently developed at Météo-France ('B21', unpublished but used e.g. in Dick et al., 2023).'

- L. 287: This is somewhat subjective and I think the sentence would be stronger if you cited the quantitative metrics here. We will list the bias values which are currently included in the Supplementary Material, only.

- L. 289: The phrase "not exactly recorded locations" is awkward wording. Please rephrase. We will change this to 'the locations of these pits change over the winter and are not exactly recorded'.

- L. 311: Should be "Sturm". Will be corrected.

- L. 452: Add "a" before "main". Will be added.

- L. 475-482: Can you please clarify whether blowing snow is simulated in the model or not? I think wind redistribution should be noted here as an important process for spatial variability of snow. Blowing snow redistribution is indeed not simulated by the model. Based on our experience with measurement at these sites (see, e.g., Mazzotti et al. 2020, https://doi.org/10.1029/2020WR027572), we do not expect wind-driven snow redistribution to be a major driver of variability given the relatively low wind speeds in the forest. Yet, we will include some considerations on the implications of this missing process in the discussion.

- L. 494: The sentence has awkward wording ("did not allow to evaluate"). Please rephrase. The sentence will be revisited.

- L. 506: Should be "prey" instead of "pray". This will be corrected, thank you for the catch.

- L. 518: Add "a" before "benchmark". Will be added.

- L. 520-521: The sentence begins with awkward wording. Please rephrase. The sentence will be rephrased as follows: 'This approach would enable ensemble simulations that better capture the unresolved spatial variability arising from heterogeneous canopy structure. Moreover, such ensemble simulations could also provide an estimate of model uncertainty when applied to forest sites where canopy descriptors are poorly constrained.'

- L. 531-534: Could the new snow density and snow compaction parameterizations also be impacting the snow depth overestimation?

Available evaluations of Crocus total density (Lafaysse et al. (2017); Viallon-Galinier et al. (2020); Ménard et al. 2021; https://doi.org/10.1175/BAMS-D-19-0329.1) do not exhibit a systematic bias in alpine environments suggesting that mechanical compaction is accurately parameterized in the model. However, some studies (e.g. Helfricht et al. 2018; https://doi.org/10.5194/hess-22-2655-2018) suggest that the density of new snow is overestimated by the Crocus parameterization of falling snow. This could lead to a temporary underestimation of snow depth after recent snowfalls while on the contrary our results exhibit an overall snow depth overestimation. Therefore, this parameterization is not expected to be the main reason for the overall snow depth overestimation. Recently, Wooley et al. (preprint, : https://doi.org/10.5194/egusphere-2024-1237), showed that snow density may be underestimated by the default Crocus configuration in Arctic environments due to the unsimulated impact of wind-drift induced compaction, yet wind speeds are low at our site. See also our reply to your main comment #1 above.

- L. 531: This focuses on one of the evaluations of the modeled snow depth, however, I think it is best to also acknowledge the prominent deficiencies in modeled snow depth at the Finland site in April (Figure S2). See my first major comment above. The deficiencies of the model at the site in Sodankylä will be specifically mentioned in a revised version of the manuscript, see our answer to the main comment #1 above.

**FIGURES**

- Figure 2, Figure S1, and Figure S2: Please add a scale bar. Scale bars will be added to all these figures.

- Figure 2: Please clarify in the caption what blue represents in the hemispherical photos. I believe it is in the sky portion outside the solar track but it would be helpful to state this in the caption. It is correct that blue represents the sky portion in the image. We will clarify this in the caption.

- Figure 4: I wonder if it would be useful to show a plot of mean direct beam transmissivity at each location on the transect? This could go just below the Fveg and could have similar dimensions/scale. This is not a required revision but merely a suggestion if it helps to show the shaded area in the open gap on the left side of the figure. Thank you for this suggestion, we will consider this addition to the Figures showing the transect.

- Figures 4, 5, 6, S3, S4, … : It could be helpful to add "S" on the left and "N" on the right at the top to indicate the south-to-north orientation of the transect. We will add these labels as suggested.

- Figure 7: I suggest adding a map on mean canopy transmissivity, which I suspect might aid in interpretation of the spatial patterns here. Thank you for this suggestion, a map of mean canopy transmissivity will be added to the figure.

---

## Author Comment (AC2)

**Exploring the potential of forest snow modelling at the tree- and snowpack layer scale – Response to reviewer # 2**

**General Comments**

The paper presents recent work by the authors to combine the strengths of two state-of-the-art snow models: the forest canopy representation from an intermediate complexity snow model (FSM2), and the detailed multi-layer snowpack model (Crocus). After outlining the two models and the process of combination into a new model (FSMCRO), two forested testing sites are introduced. Qualitative comparison of FSMCRO simulations is made to observations and baseline simulations with FSM2. The FSMCRO simulations are then interrogated at a series of points and transects to highlight differences in snow microstructure driven by location within the forest stand. The multi-physics ensemble capabilities of FSMCRO are used to investigate how robust the simulated spatial differences in snowpack microstructure are to model uncertainty. Finally, the variability in microstructure from a multi-physics ensemble driven with domain-average meteorology is compared to the variability in microstructure produced by a deterministic high-resolution simulation at over same domain.

The work is novel and showcases new modelling capabilities that are undoubted state-of-the-art. The processes simulated are relevant to readers of *The Cryosphere*. However, there are several areas that require further description, results or discussion:

The showcase-style manuscript, where different capabilities are presented and described, doesn't necessarily demonstrate significant advances in knowledge provided by the new system. I expected to see more examples of quantitatively analysis of the multi-dimensional data - e.g. evolution of CV of different parameters over time. As well, I expected the manuscript to begin to draw relationships between the (modelled) stratigraphy, the spatial structure, and the physical processes, to provide some hypotheses for future observational and/or modelling work.

The manuscript needs to be clearer about the link between the simulated results and reality. This could be achieved by presenting further quantitative statistics from the observations presented, as well as attempting to validate against microstructural observations. Similarly, the manuscript needs to provide more commentary on whether the patterns shown in the simulations are likely to be real or not, referring to available observational studies.

From a methodological point of view, the manuscript needs more discussion on differences between the FSM2 canopy model with what is implemented in FSMCRO, the reasons for the trade-offs, and discussion of the potential impact of these differences on the simulations. Also, while not the focus of the paper, the difficulties encountered when attempting to couple the models at the snow surface are mentioned, and it would be insightful to briefly expand on some of the issues encountered that led to the choice to develop a 0-layer model instead.

The paper should make a valuable contribution to *The Cryosphere* with revision.

We would like to thank the reviewer for the positive assessment of our work, the detailed review, and the constructive suggestions. Based on the general comments above, we plan two major additions in a revised version of a manuscript: 1) A comparison of model results with snow pit observations from Sodankylä, even though these were taken outside of our study domain and in other years than this considered here. For more details, please see our reply to reviewer 1, main comment #1; 2) Expand the current Sections 3.3. ('Horizontal variability of snowpack stratigraphy along a forest discontinuity at different points in time') and 3.4 ('Spatial patterns and fractional partitioning of snow stratigraphy from fully distributed simulations') to include more quantitative analysis and variability metrics, and comment on the underlying processes in Section 4.2 of the discussion ('new insights on the impact of canopy structure on snow stratigraphy'.

However, maintaining the proof-of-concept style of the paper remains important to us for two reasons: 1) its length is already at the higher limit for a TC article; and 2) a thorough validation against snow pit observations, suggestions for model improvements, and a full analysis of all dimensions of variability

covered by the simulations shall provide material for a separate study. While we agree that all these aspects are interesting, they cannot all be covered by one article. Note, for instance, recent work by Bouchard et al., who dedicated one paper separately to each of these aspects using the model SNOWPACK. In general, attempts of validation at snow pits are very rare, extremely challenging, and so far consisted of targeted efforts for existing rather than new models (e.g. Calonne et al. 2020, https://doi.org/10.5194/tc-14-1829-2020; Leppäenen et al. 2017, https://doi.org/10.3189/2015JoG14J026 ). The goal of this manuscript here is (and should remain) to bring together detailed canopy and snowpack representation for the first time, and to demonstrate that snow stratigraphy is sensitive to this heterogeneous forest structure. To our knowledge, this has never been done before and we believe it provides sufficient content for one article.

**Specific Comments**

Ln 108 - either in the methods of discussion section, it would be useful to reflect on how much the results depend on the specific model choices, noting that there are some subjective choices here.

This is a valid point, and we will comment on the choice of the two models and potential impacts on the result in Section 4.2 of the discussion ('new insights on the impact of canopy structure on snow stratigraphy'). In terms of canopy representation, important model choices are parameters related to interception and unloading as well as the choice of radiative transfer schemes. The suitability of these model choices was demonstrated in the process-level validation by Mazzotti et al. (2021). For the snow representation, current microstructure-resolving model applications basically rely on either Crocus or SNOWPACK. Uncertainty in process representation choices in Crocus are captured by using the ensemble framework, which makes the model an adequate choice for our study.

Ln 148 – perhaps add "hereafter referred to as "FSM2" after (FSM2.0.3) to distinguish the enhanced canopy model from the standard FSM2 model – see next comment.

Will be added for clarity, thank you for catching this.

Ln 159 – "The model has so far been used for research purposes" - the original FSM2 has been used in many research and operational applications - make it clear you mean the canopy version here.

This will be clarified. Please note that the 'original FSM2' does have a canopy implementation as well, which is however unsuitable for meter-resolution simulations.

Ln 205 – please provide a short commentary in the methods section on which methods remain the same as FSM2 and the extent to which others have been modified. A table would be a very handy reference for the reader.

Thank you for this suggestion, we will add a table summarizing similarities and differences to the original implementation in Appendix A1 (following the current equations).

Ln 242 – 254 – it would help the reader if the numbering and ordering aligned with the order that results are presented.

We will rearrange the order of the presented simulations to match the presentation of results, note however that this will require splitting the current point 3 into two separate points.

Ln 267-272 – this largely repeats the preceding section (2.2.3) and could be removed or combined with the above.

We agree; this paragraph will be shortened to avoid redundancy.

Ln 283 – it would be useful to report some basic quantitative stats from the validation here (e.g., overall bias, RMSE, R, CV) to give confidence in this application.

We will add overall bias values in the main article, please note that the detailed quantitative stats are already available and referenced in the Supplementary Material. We would like to keep most of the validation in the Supplementary Material to avoid distracting from what we consider to be the main aspects of this study (Sections 3.3 and 3.4)

Ln 290 – while it is understandable that the irregular and uncertain location of snowpit observations may limit a full quantitative evaluation of the FSMCRO simulations, it would be instructive to present some of the observations here if only to highlight the shortcomings of the available observations, motivate hypotheses that could be interrogated with FSMCRO and comment on how these may be validated with new observations. Not including observation of microstructure substantially reduces the readers confidence that new model system is simulating real patterns.

Some comparison to snow pit data will be included and discussed in the Supplementary Material of a revised version of the manuscript. Note however, that snow pit data is only available starting WY 2019, and related comparisons are problematic for many reasons (see your major comment above).

Ln 305 – "formation of surface melt forms/crusts (red) happens ca. 10 days earlier under-canopy than in the canopy gap" – this is not immediately clear from the figure – please add the dates to show the specific period intended.

Dates will be added.

Ln 343 – here and elsewhere (including Ln 377 and figures) it would be easier for readers in both northern and southern hemispheres if 'sun-exposed edge' and 'shaded edge' were used in place of 'south-exposed edge' and 'north-exposed edge'. Either way, please be consistent throughout the text and figures with the terms used (e.g., next sentence has 'sun-exposed edge', figures have 'n-facing').

This is a good point. We will replace instances of 'south' and 'north' with 'sun-exposed' and 'shaded' throughout the manuscript.

Ln 351 – "does not overestimate the variability of snow stratigraphy." please be specific - do you mean that in the accumulation season, vertical variability is large, whereas in the ablation season, horizontal variability is large? If so, please state this.

Yes, this is the case, and it is already stated a few lines further up. With this sentence, we meant that the qualitative differences depicted by the discrete/categorical grain type parameter are not giving an unrealistic picture of variability, as the more quantitative/continuous parameter SSA backs up the variability patterns seen in the grain type plots. We will rephrase the sentence for it to be clearer.

Ln 363 – "In contrast, snow depth variability between ensemble members at each location is in the same order of magnitude as differences between the two locations." does this mean that the structural differences are more likely real? and that the snow depth differences are not? or just that the model behaves in the same way for the same forcing? Please comment.

This result implies that prediction of structural differences is more robust than the prediction of snow depth differences, because all ensemble members agree on the structural differences (surface crust yes vs. no), while the difference in snow depth between the two locations is within the uncertainty represented by the ensemble simulation at each location. We will rephrase this to make our point clearer.

Ln 383 – "This finding provides strong evidence of the substantial impact of canopy structural heterogeneity on modelled snow stratigraphy, suggesting that the resulting variability by far exceeds model uncertainty." - was the ensemble system was validated against forested as well as open-site locations? this would be needed to conclude that model uncertainty is fully captured by the ensemble, and thus that model uncertainty is less than the explicitly resolved spatial variability.

ESCROC represents uncertainty in surface and internal snow processes and has been evaluated in a large range of environments and climatic conditions (see Lafaysse 2023; https://theses.hal.science/tel-04130109/). We therefore have confidence that model uncertainty is well represented even when near-surface atmospheric conditions are modified to account for the effect of canopy, as done in this study. Obviously, ESCROC does not represent forest-snow interactions uncertainties, but the above statement only links the impact of forest processes (and their variability) to the uncertainty in snow process representation, which is therefore appropriate.

Ln 395 - is there indirect ways to validate these sorts of results - e.g. surface temperature from thermal imaging?

Thermal imaging, especially from drone-based platforms, would certainly provide interesting datasets for validation, yet such datasets were not available within the context of this study. Note that the use of such datasets for validation is not straightforward due to the strong temporal dynamics of surface temperature (one image provides a temporal snapshot, while model forcing data was available at hourly resolution). We will add a comment on potential validation approaches as suggestion for future work in a revised version of the discussion (Section 4.4).

Ln 414 – "snowfall events" – please give dates or use annotations on figure to highlight period being referred to.

Snowfall periods will be marked in the figures.

Ln 415 "co-exist at the surface" – again please be specific about what periods are being referred to.

The period will be specified.

Ln 415 "The ensemble thus does not capture variable metamorphism rates that are tightly linked to specific canopy structure" – would we expect it to? Please comment.

No, we do not expect it to, because the variability in canopy structure is (currently) not included in the ensemble. This is why hyper-resolution simulations as shown here provide added value, and it is also the reason why we suggest that canopy processes should be added in an ensemble if uncertainty coming from canopy structure metrics is to be accounted for. We will rephrase the relevant paragraph to make this point clearer.

Ln 435-438 – a description of the issues encountered when trying to fully couple the models is not provided, so it is hard to assess how the result of Cristea et al.2022 are relevant here, and what we are to learn from these comments. Do you mean that these pitfalls should be documented in future studies? Or that this justifies your choice of a zero-layer approach? Or that Crocus is free from these issues. Please revise.

The 'numerical pitfalls' are mainly related to instabilities in surface temperatures that are common in models that allow thin layering. The results of Cristea et al. (2022) are relevant in this context because they showed that the mere choice of the number of layers in a snow model had a strong impact on model results, which does point to such numerical issues. In the case of canopy coupling, instabilities happen because the surface temperature is part of both the canopy and the snow energy balance, and the two are solved sequentially rather than in a fully-coupled way in FSM2 (as is the case in most other Land Surface Models as well). Choosing a zero-layer approach allowed us to avoid running into this issue, because thermal diffusion and any updates to snow temperatures are only treated in the snow routine. We will detail this reasoning in a revised version of the manuscript.

Ln 444 – please briefly outline the reasons provided by Nousa et al, (2023) in the text.

Nousu et al. highlighted the example of the turbulent exchange parametrizations available in ESCROC, where one of them (Martin and Lejeune 1998) applies a stability correction term that is not implemented in

MEB and therefore is not readily applicable to forest simulations done with MEB-Crocus (coupled canopy). We will mention this example in the text.

Ln 452 – further discussion on the limitations of the system is needed. Especially the potential impacts of trade-offs made in the zero-layer implementation (e.g. non-interactive canopy temperature – snow surface temperature – snowpack feedback).

We will expand this discussion, referring to studies that have shown that air temperature is a reasonable proxy of canopy temperature in most situations and discussing the implications of this assumption. We will further refer to other forest-snow models that do not account for snow surface temperature – canopy feedbacks (e.g. SNOWPACK).

Ln 464 – expected more analysis of how to quantitatively assess multi-dimensional data - e.g. evolution of CV of different parameters over time.

We will provide an example of quantitative analysis of variability, see our reply to your general comments above. Note, however, that a thorough analysis of the multi-dimensional data is beyond the scope of this study given the paper's current length.

Ln 464 – expected the manuscript to begin to draw relationships between the (modelled) stratigraphy, the spatial structure, and the physical processes, to provide some hypotheses for future observational and/or modelling work.

Some discussion of the relationship between modelled stratigraphy and spatial canopy structure is included in the subsequent paragraphs (465ff). We will expand the discussion of relationships between modelled stratigraphy and the physical processes in this paragraph and include hypotheses for future work in Section 4.4.

Ln 473 - "The second mechanism [insolation-microstructure relationships] has, to our knowledge, not been captured by any simulation prior to this study."- please comment on whether these patterns have been observed, and if so, provide references.

To our knowledge, no observational study targeting how snow microstructure varies with insolation patterns in discontinuous forest exists. We will mention this explicitly in the revised discussion and highlight how hyper-resolution simulations with FSMCRO can be useful for identifying processes (and process interactions) that have never been studied.

Ln 577 – as per comment on Ln 205 – would be useful either here or in methods section to have a list or table of which schemes are identical, which are slightly, modified and which required larger modification.

See our answer to your comment above, a table will be added to the Appendix.

**Technical corrections**

Ln 129 – please provide a reference for the standalone version of Crocus.

The standalone version of Crocus is still unpublished but will be included in a publication covering the latest developments of Crocus (Lafaysse et al., in preparation). The standalone version of the code is currently available at https://opensource.umr-cnrm.fr/projects/surfex_git2/wiki/Install_standalone_version_of_Crocus. Yet, because a migration to GitHub is planned in the next few months, including this link to the publication is obsolete. We will ensure that the correct link is listed and regularly updated in the GitHub repository of FSMCRO.

Ln 130 – please explain what SVS2 is, as the reference (Vionnet et al, 2022) does not mention it.

SVS2 is the Land Surface Model used at Environment and Climate Change Canada, we will specify this in the revised manuscript. In Vionnet et al. (2022), the model is presented in Section 2.2. The version SVS-2 including Crocus is now also used in Wooley et al. (preprint: https://doi.org/10.5194/egusphere-2024-1237), this citation will be added.

Ln 173 – add 'incoming' before 'short-'

Will be done as suggested.

Ln 288 – 'FMI' - please expand the acronym.

The acronym was already spelled out in line 217 but not defined, this will be corrected, thank you for the catch.

Ln 299 – 'SSA' please expand first use of acronym.

Will be done, thanks for catching this.

Ln 315 – "discontinuous forest transect" -> "transect through discontinuous forest". The former implies the transect is discontinuous.

Will be modified as suggested.

Ln 323 – "peak of winter" -> "peak winter accumulation"

We will adapt this and check other occurrences of 'peak of winter' throughout the manuscript.

---

## Author Response (AR1)

**Exploring the potential of forest snow modelling at the tree- and snowpack layer scale – Response to reviewer # 1**

**Summary and recommendation**

Mazotti et al. develop and present a new physics-based, multi-layer, hyper-resolution snow model (FSMCRO) that can represent high spatial and vertical resolution snow properties including grain type, density, temperature, and other snow parameters. This was achieved through a one way coupling between the FSM2 canopy model and the ensemble Crocus model, with the added benefit that ensemble simulations provides a mean for assessing uncertainty. The paper focuses on introducing and demonstrating the model at two well studied snow sites (Finland and Switzerland), with only qualitative validation ("plausibility"). The model shows reasonable representation of snow depth patterns in Switzerland (focus in the main paper) but less so in Finland (supp. material). Overall, the model shows realistic spatial variations in key snow properties (grain size, SSA) and their evolution in time along a transect spanning a forest gap with variable radiation and interception dynamics. Through the use of the ensembles and spatial simulations, the study also finds that snowpack variability (due to canopy effects on snow processes) is more important than model uncertainty.

Overall, I find this paper potentially offers a significant advance in our ability to resolve very localized snow properties which will be of interest and use to research in snow-forest interactions, wildlife ecology, and possibly avalanche studies. I think the scientific and presentation are generally of high quality, though I offer some comments and suggestion for further improvement. My main concern is about the minimal validation effort and the apparent deficiencies in snow depth simulation at one of the sites (See #1 below), and therefore request the authors consider these before publication. I emphasize this paper should be published following attention to these comments.

We would like to thank the reviewer for the careful review, the overall positive assessment of our work, and the constructive comments. Please find our answers to each comment below in blue. Line numbers correspond to revised manuscript with tracked changes.

**MAIN COMMENTS**

1.       While the paper does not present a detailed validation but rather a demonstration of the new model, it seems there is still an opportunity to provide additional analysis to understand the "plausibility" of the model and needs for future improvements. For instance, the paper references weekly snow pit data at the Finland site, but does not make use of them due to issues with geolocation. I would argue that the geolocation issue with the pits does not preclude such a comparison, as multiple location from the domain could be selected, along with the ensemble members in order to understand the range of possible snow profiles simulated by FSMCRO. I think that a comparison between the FSMCRO ensemble and the snow pit data (grain type, density, etc.) could still be informative, even if done on a qualitative basis given the recognized challenges in comparing multi-layer snow models to snow pits. This might help to identify the plausibility of the model as well as possible deficiencies and areas for future development in the model. At the same time, this may require attention to the prominent errors in FSMCRO snow depth that are apparent at the Finland site (Figure S2, where even normalized snow depths are quite different from observations). As noted by the authors: "an adequate reproduction of observed snow depth patterns is a prerequisite for a meaningful subsequent analysis of snowpack vertical properties" (L. 285-286). Comparing to the Finland snowpit data might be helpful for diagnosing possible reasons for the deficient snow depth representation (e.g., bulk snow density?).

Indeed, validation should not be a major component of this study, yet we understand the reviewer's point and have further investigated the two issues mentioned. Regarding the relative snow depth patterns in Sodankylä (Fig S2), we have identified two reasons that have contributed to the differences in simulation between FSMCRO and FSM2: 1) For both simulations we have used a precipitation undercatch correction that was determined based on FSM2 simulations at the open site. This is a standard procedure in case of FSM (e.g., Essery et al. 2017, https://doi.org/10.5194/gi-5-219-2016), but not for Crocus (e.g., Nousu et al., 2023, https://doi.org/10.5194/egusphere-2023-338). Since this work was intended as a first model demonstration, with thorough model validation and tuning to follow separately, we left this issue unattended. However, we

repeated the simulations of FSMCRO without undercatch correction, which improves the match with observations (see Figure below, lower right panel); 2) The ablation rates of FSMCRO are somewhat lower compared to FSM2, again, owing to the lack of any specific model tuning of FSMCRO at this point. Consequently, snow distribution patterns that better match those observed are only attained a few days later (see Figure below, upper right panel). The below-canopy albedo or the subcanopy turbulent exchange parameters may require some tuning to fix this issue, as noted in the discussion. We have replaced the FSMCRO simulation in Figure S2 by the simulation without undercatch, specifying our reasoning in the text (Supplementary Material S1).

[Figure]

Regarding a comparison of FSMCRO simulations with snow pit data, we believe this deserves full attention in a follow up paper, as shown by various examples of dedicated studies involving snow pit validations (Bouchard et al., 2024: https://doi.org/10.5194/tc-18-2783-2024; Calonne et al. 2020, https://doi.org/10.5194/tc-14-1829-2020; Leppänen et al. 2017, https://doi.org/10.3189/2015JoG14J026 ). Yet, to address your comment we explored first test simulations, which are now presented and discussed in a new section of the Supplementary Material (S2). Note however that forest snow pits were only available since WY 2019, which does not correspond to the study period considered in the paper. We therefore had to perform simulations for additional years at a range of locations that appear to be qualitatively 'similar' to the area where the snow pits were located (which is outside of our study area). Comparison of these simulations with snow depths recorded in the snow pits is shown in Figure S3, while exemplary comparisons of FSMCRO simulations with snow pit observations are shown as well, including both a seasonal deterministic simulation (Figure S4) as well as ensemble outputs for two dates (Figure S5). A comparison of grain type however was beyond the scope of this additional analysis and shall be left for a dedicated follow-up study. As noted by Leppänen et al. (2016; https://doi.org/10.5194/gi-5-163-2016), these measurements are most prone to observer-related biases. Moreover, the frequent use of subclasses and mixed grain types as well as the conversion to a quantitative metric would require substantial additional quality control and postprocessing.

2.      Several figures in the paper are not readable for someone with a red-green vision deficiency. As such, those readers may not be able to distinguish (for instance) the different snow grain types (e.g., melt forms vs. precipitation particles). I recognize this is not the fault of the authors as they are following the conventions from the Fierz et al. (2009) international snow classification report. However, I would suggest the authors consider whether something can be done to help these readers (e.g., adding a small hatch pattern to the green colors).

Thank you for pointing this out, we recognize that our figures showing grain types are not adapted to red-green vision deficiency. However, this colormap is an international convention (see Fierz et al. (2009)) that required substantial work to be established and is now widely accepted in the community. We believe it is important to be consistent with this convention and have therefore not changed the figures in the main article.

I recommend adding snow hardness and snow liquid water content (LWC) as new figures in the supplement (similar to Figures S3-S4), as the capability for mapping these variables spatially may be of high interest to other researchers. The paper references wildlife ecology, and for that the snow hardness is a relevant parameter. Likewise, snowmelt studies and microwave remote sensing (e.g., GPR) may benefit from a model that can resolve spatial variations in LWC.

We have added figures showing liquid water content and ram resistance to the Supplementary Material (former S2, now S3) and refer to it in the main text (L369), thank you for the suggestion.

**Line Comments**

**-** L. 27: This should be "tools".

Thank you for the catch, this has been corrected.

- L. 30-35: The opening sentence is rather long and cumbersome. I recommend breaking it into two or more sentences.

The sentence has been broken down into five shorter sentences, see L30-36.

- L. 215: Add "an" after "as".

Has been added.

- L. 256: The sentence begins with awkward wording. Please rephrase.

The beginning of the sentence has rephrased, see L267ff.

- L. 287: This is somewhat subjective and I think the sentence would be stronger if you cited the quantitative metrics here.

We have added bias values which were formerly in the Supplementary Material (L 300).

- L. 289: The phrase "not exactly recorded locations" is awkward wording. Please rephrase.

This sentence has been removed in the revised version of the manuscript. Note that similar (reworded) content is now in Supplementary Material S2.

- L. 311: Should be "Sturm".

Has been corrected, thank you for the catch.

- L. 452: Add "a" before "main".

Has been added.

- L. 475-482: Can you please clarify whether blowing snow is simulated in the model or not? I think wind redistribution should be noted here as an important process for spatial variability of snow.

Blowing snow redistribution is indeed not simulated by the model, this is now explicitly mentioned in the model description section of the Appendix (A1). Based on our experience with measurement at these sites (see, e.g., Mazzotti et al. 2020, https://doi.org/10.1029/2020WR027572) we do not expect wind-driven snow redistribution to be a major driver of variability given the relatively low wind speeds in the forest. This is now noted as well (L670).

- L. 494: The sentence has awkward wording ("did not allow to evaluate"). Please rephrase.

The sentence has been revisited (L531ff).

- L. 506: Should be "prey" instead of "pray".

This has been corrected, thank you for the catch.

- L. 518: Add "a" before "benchmark".

Has been added.

- L. 520-521: The sentence begins with awkward wording. Please rephrase.

The sentence has been rephrased, see L570ff.

- L. 531-534: Could the new snow density and snow compaction parameterizations also be impacting the snow depth overestimation?

Available evaluations of Crocus total density (Lafaysse et al. (2017); Viallon-Galinier et al. (2020); Ménard et al. 2021; https://doi.org/10.1175/BAMS-D-19-0329.1) do not exhibit a systematic bias in alpine environments suggesting that mechanical compaction is accurately parameterized in the model. However, some studies (e.g. Helfricht et al. 2018; https://doi.org/10.5194/hess-22-2655-2018) suggest that the density of new snow is overestimated by the Crocus parameterization of falling snow. This could lead to a temporary underestimation of snow depth after recent snowfalls while on the contrary our results exhibit an overall snow depth overestimation. Therefore, this parameterization is not expected to be the main reason for the overall snow depth overestimation. Recently, Wooley et al. (preprint, : https://doi.org/10.5194/egusphere-2024-1237), showed that snow density may be underestimated by the default Crocus configuration in Arctic environments due to the unsimulated impact of wind-drift induced compaction, yet wind speeds are low at our site. See also our reply to your main comment #1 above.

- L. 531: This focuses on one of the evaluations of the modeled snow depth, however, I think it is best to also acknowledge the prominent deficiencies in modeled snow depth at the Finland site in April (Figure S2). See my first major comment above.

The model simulations were updated, and the performance at Sodankylä is now addressed more specifically in the Supplementary Material S1, see our answer to the main comment #1 above.

**FIGURES**

- Figure 2, Figure S1, and Figure S2: Please add a scale bar.

Scale bars have been added to all these figures.

- Figure 2: Please clarify in the caption what blue represents in the hemispherical photos. I believe it is in the sky portion outside the solar track but it would be helpful to state this in the caption.

It is correct that blue represents the sky portion in the image, it is now clarified in the caption.

- Figure 4: I wonder if it would be useful to show a plot of mean direct beam transmissivity at each location on the transect? This could go just below the Fveg and could have similar dimensions/scale. This is not a required revision but merely a suggestion if it helps to show the shaded area in the open gap on the left side of the figure.

Thank you for this suggestion, we have added a bar with mean direct beam transmissivity to all Figures showing the transect.

- Figures 4, 5, 6, S3, S4, … : It could be helpful to add "S" on the left and "N" on the right at the top to indicate the south-to-north orientation of the transect.

We have added these labels as suggested.

- Figure 7: I suggest adding a map on mean canopy transmissivity, which I suspect might aid in interpretation of the spatial patterns here.

Thank you for this suggestion, maps of mean canopy transmissivity have been added in the Supplementary Material S4 for reference. Together with the canopy height model from Figure 1, this will aid interpretation of the patterns.

**Response to reviewer # 2**

**General Comments**

The paper presents recent work by the authors to combine the strengths of two state-of-the-art snow models: the forest canopy representation from an intermediate complexity snow model (FSM2), and the detailed multi-layer snowpack model (Crocus). After outlining the two models and the process of combination into a new model (FSMCRO), two forested testing sites are introduced. Qualitative comparison of FSMCRO simulations is made to observations and baseline simulations with FSM2. The FSMCRO simulations are then interrogated at a series of points and transects to highlight differences in snow microstructure driven by location within the forest stand. The multi-physics ensemble capabilities of FSMCRO are used to investigate how robust the simulated spatial differences in snowpack microstructure are to model uncertainty. Finally, the variability in microstructure from a multi-physics ensemble driven with domain-average meteorology is compared to the variability in microstructure produced by a deterministic high-resolution simulation at over same domain.

The work is novel and showcases new modelling capabilities that are undoubted state-of-the-art. The processes simulated are relevant to readers of *The Cryosphere*. However, there are several areas that require further description, results or discussion:

The showcase-style manuscript, where different capabilities are presented and described, doesn't necessarily demonstrate significant advances in knowledge provided by the new system. I expected to see more examples of quantitatively analysis of the multi-dimensional data - e.g. evolution of CV of different parameters over time. As well, I expected the manuscript to begin to draw relationships between the (modelled) stratigraphy, the spatial structure, and the physical processes, to provide some hypotheses for future observational and/or modelling work.

The manuscript needs to be clearer about the link between the simulated results and reality. This could be achieved by presenting further quantitative statistics from the observations presented, as well as attempting to validate against microstructural observations. Similarly, the manuscript needs to provide more commentary on whether the patterns shown in the simulations are likely to be real or not, referring to available observational studies.

From a methodological point of view, the manuscript needs more discussion on differences between the FSM2 canopy model with what is implemented in FSMCRO, the reasons for the trade-offs, and discussion of the potential impact of these differences on the simulations. Also, while not the focus of the paper, the difficulties encountered when attempting to couple the models at the snow surface are mentioned, and it would be insightful to briefly expand on some of the issues encountered that led to the choice to develop a 0-layer model instead.

The paper should make a valuable contribution to *The Cryosphere* with revision.

We would like to thank the reviewer for the positive assessment of our work, the detailed review, and the constructive suggestions. Please find our answers to each comment below in blue. Line numbers correspond to revised manuscript with tracked changes.

Based on the general comments above, we have made the following major additions in the revised version of our manuscript: 1) We have added a comparison of model results with snow pit observations from Sodankylä, even though these were only available from outside of our study domain and in other years than this considered here (see new Section S2 with three new figures). 2) We have further redone the existing simulations for Sodankylä with adaptations to address your concerns about the degraded performance of FSMCRO at this site, which is now mostly mitigated (see new Figure S2). 3) We have added performance metrics in the main article and greatly extended the Supplementary material.

However, maintaining the proof-of-concept style of the paper remains important to us for two reasons: 1) its length is already at the higher limit for a TC article; and 2) a thorough validation against snow pit

observations, suggestions for model improvements, and a full analysis of all dimensions of variability covered by the simulations shall provide material for a separate study. While we agree that all these aspects are interesting, they cannot all be covered by one article. Note, for instance, recent work by Bouchard et al., who dedicated one paper separately to each of these aspects using the model SNOWPACK. In general, attempts of validation at snow pits are very rare, extremely challenging, and so far consisted of targeted efforts for existing rather than new models (e.g. Calonne et al. 2020, https://doi.org/10.5194/tc-14-1829-2020; Leppäenen et al. 2017, https://doi.org/10.3189/2015JoG14J026 ). The goal of this manuscript here is (and should remain) to bring together detailed canopy and snowpack representation for the first time, and to demonstrate that snow stratigraphy is sensitive to this heterogeneous forest structure. To our knowledge, this has never been done before and we believe it provides sufficient content for one article. Finally, please also see our response to main comment #1 from reviewer 1.

**Specific Comments**

Ln 108 - either in the methods of discussion section, it would be useful to reflect on how much the results depend on the specific model choices, noting that there are some subjective choices here.

This is a valid point, and we have justified the choice of these two models in the respective sections of the methods (L124ff, L164ff), and related limitations are acknowledged in the discussion (L530ff). In terms of canopy representation, important model choices are parameters related to interception and unloading as well as the choice of radiative transfer schemes. The suitability of these model choices was demonstrated in the process-level validation by Mazzotti et al. (2020b). For the snow representation, current microstructure-resolving model applications basically rely on either Crocus or SNOWPACK. Uncertainty in process representation choices in Crocus are captured by using the ensemble framework, which makes the model an adequate choice for our study (see L126ff).

Ln 148 – perhaps add "hereafter referred to as "FSM2" after (FSM2.0.3) to distinguish the enhanced canopy model from the standard FSM2 model – see next comment.

Has been added as suggested for clarity (L154).

Ln 159 – "The model has so far been used for research purposes" - the original FSM2 has been used in many research and operational applications - make it clear you mean the canopy version here.

This has been clarified (L164). Please note that the 'original FSM2' does have a canopy implementation as well, which is however unsuitable for meter-resolution simulations.

Ln 205 – please provide a short commentary in the methods section on which methods remain the same as FSM2 and the extent to which others have been modified. A table would be a very handy reference for the reader.

Thank you for this suggestion, we have added a table summarizing similarities and differences to the original implementation in Appendix A1 (Table A1).

Ln 242 – 254 – it would help the reader if the numbering and ordering aligned with the order that results are presented.

We have rearranged the order of the presented simulations to match the presentation of results, which required splitting the current point 3 into two separate points (L250ff).

Ln 267-272 – this largely repeats the preceding section (2.2.3) and could be removed or combined with the above.

We have shortened the paragraph to avoid redundancy (L275ff).

Ln 283 – it would be useful to report some basic quantitative stats from the validation here (e.g., overall bias, RMSE, R, CV) to give confidence in this application.

We have added overall bias and CV values in the main article (L295ff), please note that the detailed quantitative statistics are already available and referenced in the Supplementary Material. We have kept most of the validation in the Supplementary Material to avoid distracting from what we consider to be the main aspects of this study (Sections 3.3 and 3.4)

Ln 290 – while it is understandable that the irregular and uncertain location of snowpit observations may limit a full quantitative evaluation of the FSMCRO simulations, it would be instructive to present some of the observations here if only to highlight the shortcomings of the available observations, motivate hypotheses that could be interrogated with FSMCRO and comment on how these may be validated with new observations. Not including observation of microstructure substantially reduces the readers confidence that new model system is simulating real patterns.

A comparison to snow pit data has been included and discussed in a new section of the Supplementary Material of a revised version of the manuscript (S3). Note however, that snow pit data were only available starting WY 2019 and outside of our study domain (see reply to your general comments above).

Ln 305 – "formation of surface melt forms/crusts (red) happens ca. 10 days earlier under-canopy than in the canopy gap" – this is not immediately clear from the figure – please add the dates to show the specific period intended.

Dates have been added (L321).

Ln 343 – here and elsewhere (including Ln 377 and figures) it would be easier for readers in both northern and southern hemispheres if 'sun-exposed edge' and 'shaded edge' were used in place of 'south-exposed edge' and 'north-exposed edge'. Either way, please be consistent throughout the text and figures with the terms used (e.g., next sentence has 'sun-exposed edge', figures have 'n-facing').

This is a good point. We have replaced or complemented instances of 'south' and 'north' with 'sun-exposed' and 'shaded' and checked our wording for consistency throughout the manuscript.

Ln 351 – "does not overestimate the variability of snow stratigraphy." please be specific - do you mean that in the accumulation season, vertical variability is large, whereas in the ablation season, horizontal variability is large? If so, please state this.

Yes, this is the case, and it is already stated a few lines further up. With this sentence, we meant that the qualitative differences depicted by the discrete/categorical grain type parameter are not giving an unrealistic picture of variability, as the more quantitative/continuous parameter SSA backs up the variability patterns seen in the grain type plots, which is also already stated. We thus removed the sentence to avoid confusion.

Ln 363 – "In contrast, snow depth variability between ensemble members at each location is in the same order of magnitude as differences between the two locations." does this mean that the structural differences are more likely real? and that the snow depth differences are not? or just that the model behaves in the same way for the same forcing? Please comment.

This result implies that prediction of structural differences is more robust than the prediction of snow depth differences, because all ensemble members agree on the structural differences (surface crust yes vs. no), while the difference in snow depth between the two locations is within the uncertainty represented by the ensemble simulation at each location. We have added this explanation to make our point clearer (L383ff).

Ln 383 – "This finding provides strong evidence of the substantial impact of canopy structural heterogeneity on modelled snow stratigraphy, suggesting that the resulting variability by far exceeds model uncertainty." - was the ensemble system was validated against forested as well as open-site locations? this would be needed

to conclude that model uncertainty is fully captured by the ensemble, and thus that model uncertainty is less than the explicitly resolved spatial variability.

ESCROC represents uncertainty in surface and internal snow processes and has been evaluated in a large range of environments and climatic conditions (see Lafaysse 2023; https://theses.hal.science/tel-04130109/). We therefore have confidence that model uncertainty is well represented even when near-surface atmospheric conditions are modified to account for the effect of canopy, as done in this study. This is now specifically stated in the methods section (L216ff). Obviously, ESCROC does not represent forest-snow interactions uncertainties, but the above statement only links the impact of forest processes (and their variability) to the uncertainty in snow process representation, which is therefore appropriate.

Ln 395 - is there indirect ways to validate these sorts of results - e.g. surface temperature from thermal imaging?

Thermal imaging, especially from drone-based platforms, would certainly provide interesting datasets for validation, yet such datasets were not available within the context of this study. Note that the use of such datasets for validation is not straightforward due to the strong temporal dynamics of surface temperature (one image provides a temporal snapshot, while model forcing data was available at hourly resolution). We now comment on this potential avenue for future research in Section 4.4 (L605 ff.).

Ln 414 – "snowfall events" – please give dates or use annotations on figure to highlight period being referred to.

The two snowfall events are now referred to in the caption of Figure 8.

Ln 415 "co-exist at the surface" – again please be specific about what periods are being referred to.

The period (Feb 12th to 22nd) has been specified (L437).

Ln 415 "The ensemble thus does not capture variable metamorphism rates that are tightly linked to specific canopy structure" – would we expect it to? Please comment.

No, we do not expect it to, because the variability in canopy structure is (currently) not included in the ensemble. This is now stated specifically (L437). This is also why hyper-resolution simulations as shown here provide added value, and it the reason why we suggest that canopy processes should be added in an ensemble if uncertainty coming from canopy structure metrics is to be accounted for. We have expanded the relevant paragraph to make this point cleared (L566ff).

Ln 435-438 – a description of the issues encountered when trying to fully couple the models is not provided, so it is hard to assess how the result of Cristea et al.2022 are relevant here, and what we are to learn from these comments. Do you mean that these pitfalls should be documented in future studies? Or that this justifies your choice of a zero-layer approach? Or that Crocus is free from these issues. Please revise.

The 'numerical pitfalls' are mainly related to instabilities in surface temperatures that are common in models that allow thin layering. The results of Cristea et al. (2022) are relevant in this context because they showed that the mere choice of the number of layers in a snow model had a strong impact on model results, which does point to such numerical issues. We now mention this explicitly (L461). In the case of canopy coupling, instabilities happen because the surface temperature is part of both the canopy and the snow energy balance, and the two are solved sequentially rather than in a fully-coupled way in FSM2 (as is the case in most other Land Surface Models as well). This is already stated in the Section 2.1.3 (L185ff). Choosing a zero-layer approach allowed us to avoid running into this issue, because thermal diffusion and any updates to snow temperatures are only treated in the snow routine. This is now detailed in the discussion (L466ff).

Ln 444 – please briefly outline the reasons provided by Nousa et al, (2023) in the text.

Nousu et al. highlighted the example of the turbulent exchange parametrizations available in ESCROC, where one of them (Martin and Lejeune 1998) applies a stability correction term that is not implemented in MEB and therefore is not readily applicable to forest simulations done with MEB-Crocus (coupled canopy). This has been added to the text (L469ff).

Ln 452 – further discussion on the limitations of the system is needed. Especially the potential impacts of trade-offs made in the zero-layer implementation (e.g. non-interactive canopy temperature – snow surface temperature – snowpack feedback).

We have expanded this discussion, referring to studies that have shown that air temperature is a reasonable proxy of canopy temperature in most situations and discussing the implications of this assumption. We further refer to other forest-snow models that do not account for snow surface temperature – canopy feedbacks (e.g. SNOWPACK). See L479ff.

Ln 464 – expected more analysis of how to quantitatively assess multi-dimensional data - e.g. evolution of CV of different parameters over time.

Please refer to our response to the general comments above.

Ln 464 – expected the manuscript to begin to draw relationships between the (modelled) stratigraphy, the spatial structure, and the physical processes, to provide some hypotheses for future observational and/or modelling work.

Some discussion of the relationship between modelled stratigraphy, the spatial canopy structure, and the underlying processes, is already included in the subsequent paragraphs (500ff). We have further formulated a research question targeting these relationships that could be explored in future work (L561ff).

Ln 473 - "The second mechanism [insolation-microstructure relationships] has, to our knowledge, not been captured by any simulation prior to this study."- please comment on whether these patterns have been observed, and if so, provide references.

To our knowledge, no observational study targeting how snow microstructure varies with insolation patterns in discontinuous forest exists. We have mentioned this explicitly in the revised discussion (L539ff) and highlight how hyper-resolution simulations with FSMCRO can be useful for identifying processes (and process interactions) that have never been studied (L560).

Ln 577 – as per comment on Ln 205 – would be useful either here or in methods section to have a list or table of which schemes are identical, which are slightly, modified and which required larger modification.

See our answer to your comment above, a table has been added to the Appendix A1.

**Technical corrections**

Ln 129 – please provide a reference for the standalone version of Crocus.

The standalone version of Crocus is still unpublished but will be included in a publication covering the latest developments of Crocus (Lafaysse et al., in preparation). The standalone version of the code is currently available at https://opensource.umr-cnrm.fr/projects/surfex_git2/wiki/Install_standalone_version_of_Crocus. Yet, because a migration to GitHub is planned in the next few months, including this link to the publication is obsolete. We will ensure that the correct link is listed and regularly updated in the GitHub repository of FSMCRO.

Ln 130 – please explain what SVS2 is, as the reference (Vionnet et al, 2022) does not mention it.

SVS2 is the Land Surface Model used at Environment and Climate Change Canada, this is now specified (L134). In Vionnet et al. (2022), the model is presented in Section 2.2. The version SVS-2 including Crocus is now also used in Wooley et al. (preprint: https://doi.org/10.5194/egusphere-2024-1237), this citation has been added.

Ln 173 – add 'incoming' before 'short-'

Has been added as suggested (L180).

Ln 288 – 'FMI' - please expand the acronym.

The acronym was already spelled out in line 217 but not defined, this has been corrected (L224), thank you for the catch.

Ln 299 – 'SSA' please expand first use of acronym.

The acronym has been expanded at its first use (L314), thank you for catching this.

Ln 315 – "discontinuous forest transect" -> "transect through discontinuous forest". The former implies the transect is discontinuous.

Has been modified as suggested.

Ln 323 – "peak of winter" -> "peak winter accumulation"

This has been changed throughout the manuscript.